# Sexual failure decreases sweet taste perception in male *Drosophila* via dopaminergic signaling

Gaohang Wang[1], Wei Qi[2], Rui Huang[3]*, Liming Wang[1,4,5]*

[1]Institute of Molecular Physiology, Shenzhen Bay Laboratory, Shenzhen, China; [2]Department of Biology, Stanford University, Stanford, United States; [3]Guangyang Bay Laboratory, Chongqing Institute for Brain and Intelligence, Chongqing, China; [4]School of Public Health, Capital Medical University, Beijing, China; [5]Chinese Institutes for Medical Research, Beijing, China

## eLife Assessment

This study provides **valuable** findings on the effects of mating experience on sweet taste perception. The data as presented provide **convincing** evidence that the dopaminergic signaling-mediated reward system underlies this mating state-dependent behavioral modulation. The work will interest neuroscientists and particularly biologists working on neuromodulation and the effects of internal states on sensory perception.

**\*For correspondence:**
huangrui0716@sina.com (RH);
lmwang83@cimrbj.ac.cn (LW)

**Competing interest:** The authors declare that no competing interests exist.

**Abstract** Sweet taste perception, a critical aspect of the initiation of feeding behavior, is primarily regulated by an animal's internal metabolic state. However, non-metabolic factors, such as motivational and emotional states, can also influence peripheral sensory processing and hence feeding behavior. While mating experience is known to induce motivational and emotional changes, its broader impact on other innate behaviors, such as feeding, remains largely uncharacterized. In this study, we demonstrated that the mating failure of male fruit flies suppressed sweet taste perception via dopamine signaling in specific neural circuitry. Upon repetitive failure in courtship, male flies exhibited a sustained yet reversible decline of sweet taste perception, as measured by the proboscis extension reflex (PER) towards sweet tastants as well as the neuronal activity of sweet-sensing Gr5a$^+$ neurons in the proboscis. Mechanistically, we identified a small group of dopaminergic neurons projecting to the subesophageal zone (SEZ) and innervating with Gr5a$^+$ neurons as the key modulator. Repetitive sexual failure decreased the activity of these dopaminergic neurons and in turn, suppressed Gr5a$^+$ neurons via Dop1R1 and Dop2R receptors. Our findings revealed a critical role for dopaminergic signaling in integrating reproductive experience with appetitive sensory processing, providing new insights into the complex interactions between different innate behaviors and the role of brain's reward systems in regulating internal motivational and emotional states.

## Introduction

As an innate behavior, feeding is essential for the survival and reproduction of all animal species, regulated by a dynamic interplay of internal and external factors such as nutritional status, circadian rhythms, food quality, and emotional states (*Atasoy et al., 2012*; *Hergarden et al., 2012*; *Lim et al., 2012*; *Morton et al., 2006*; *Park et al., 2016*; *Shiu et al., 2022*; *Xu et al., 2008*; *Yapici et al., 2016*; *Zhan et al., 2016*). Beyond nutritive value, palatable food also serves as a powerful reward signal that regulates feeding behavior by influencing various aspects of learning, memory, and motivation

(*Dayan and Balleine, 2002*; *Wise, 2004*). The function of the dopamine system, a key regulator in reward processing, is evolutionarily conserved across species. Studies in both fruit flies and rodents demonstrate that the consumption of sugar or other highly appealing foods triggers a robust dopamine response. The neural substrates of these responses, such as the mushroom bodies in fruit flies and the ventral tegmental area in rodents, underscore the universality of this reward circuit (*Burke et al., 2012*; *Heymann et al., 2020*; *Liu et al., 2012*; *Takahashi et al., 2016*; *Yokel and Wise, 1975*).

Mating behavior, as another critical driver of the survival and reproduction of animal species, is also influenced by external stimuli and internal states (*Clowney et al., 2015*; *Kallman et al., 2015*; *Kohatsu et al., 2011*). Failure in mating behavior leads to various behavioral adaptations and even abnormalities. For example, male fruit flies exhibit courtship conditioning, where unsuccessful mating attempts reduce subsequent courtship toward inappropriate mates, demonstrating behavioral plasticity aimed at enhancing future reproductive success (*Keleman et al., 2012*). Meanwhile, the effect of mating failure extends beyond reproductive behavior. It has been shown that mating failure increases alcohol consumption in fruit flies (*Shohat-Ophir et al., 2012*), a behavior that mirrors reward-seeking tendencies such as substance abuse seen in higher organisms under stress. Recent studies also demonstrate that mating-related sensory cues can recalibrate female odor-driven preferences through a dopamine-gated learning circuit in the mushroom body (*Boehm et al., 2022*). In males, dopamine dynamically adjusts the temporal integration window of copulation decision neurons to regulate mating persistence (*Gautham et al., 2024*). Additionally, mating itself has a rewarding nature. In male flies, approaching copulation activates dopaminergic neurons that suppress visual threat responses, suggesting that the reward system can override aversive cues to promote mating behavior (*Cazalé-Debat et al., 2024*; *Zhang et al., 2016*). In male mice, after interaction with females, chemical signals are detected by the accessory olfactory bulb and transmitted to neural circuits that regulate mating behavior (*Decoster et al., 2024*). These signals are then projected to the ventral tegmental area, where serotonin neurons play a crucial role in encoding both reward acquisition and anticipation (*Hashikawa et al., 2016*).

Given the established impact of mating experience on behavioral plasticity, we aimed to investigate whether it also affected reward perception and processing in fruit flies. Specifically, we examined how courtship failure influenced sweet sensitivity and feeding behavior, both of which were tightly linked to reward. In our study, we found that sexual failure led to decreased sweet sensitivity and feeding behavior in male fruit flies, highlighting a close link between two important innate behaviors. Furthermore, we identified a small group of dopaminergic neurons that projected to the SEZ of the fly brain, modulating the activity of sweet-sensing gustatory neurons in a manner dependent on sexual experience. These dopaminergic neurons were functionally epistatic to sweet-sensing neurons and promoted behavioral responses to appetitive cues, but their activity decreased following courtship failure. As a result, the activity of $Gr5a^+$ neurons in the proboscis was reduced via signaling through Dop1R1 and Dop2R receptors, leading to reduced sweet sensitivity. These findings suggest that courtship failure inhibits the dopamine-mediated reward system in the brain, directly influencing sweet perception and feeding behavior. This provides key insights into how reproductive stressors, such as mating failure, can affect broader behavioral output, expanding our understanding of the complex interplay between instinctive drives.

## Results

### Courtship failure reduced sweet sensitivity in male flies

To investigate whether mating experience influenced feeding behavior in *Drosophila melanogaster*, we employed a modified courtship conditioning paradigm (*McBride et al., 1999*; *Shohat-Ophir et al., 2012*; *Figure 1A*) and examined three groups of males with different mating experiences *in prior*. In *Drosophila*, mated females typically refuse further copulation, leading males exposed to mated females to experience repeated sexual rejection. In contrast, males paired with multiple virgin females could engage in multiple rounds of successful copulations. In our experiments, virgin males were isolated from females for 4 days, after which they were divided into three groups on the fifth day: the Naïve group (continued isolation for 4.5 hr), the Failed group (each male exposed to 25 mated females for 4.5 hr), and the Satisfied group (each male exposed to 25 virgin females for 4.5 hr). Following the treatment, we measured sucrose intake using the MAFE (MAual FEeding) assay we

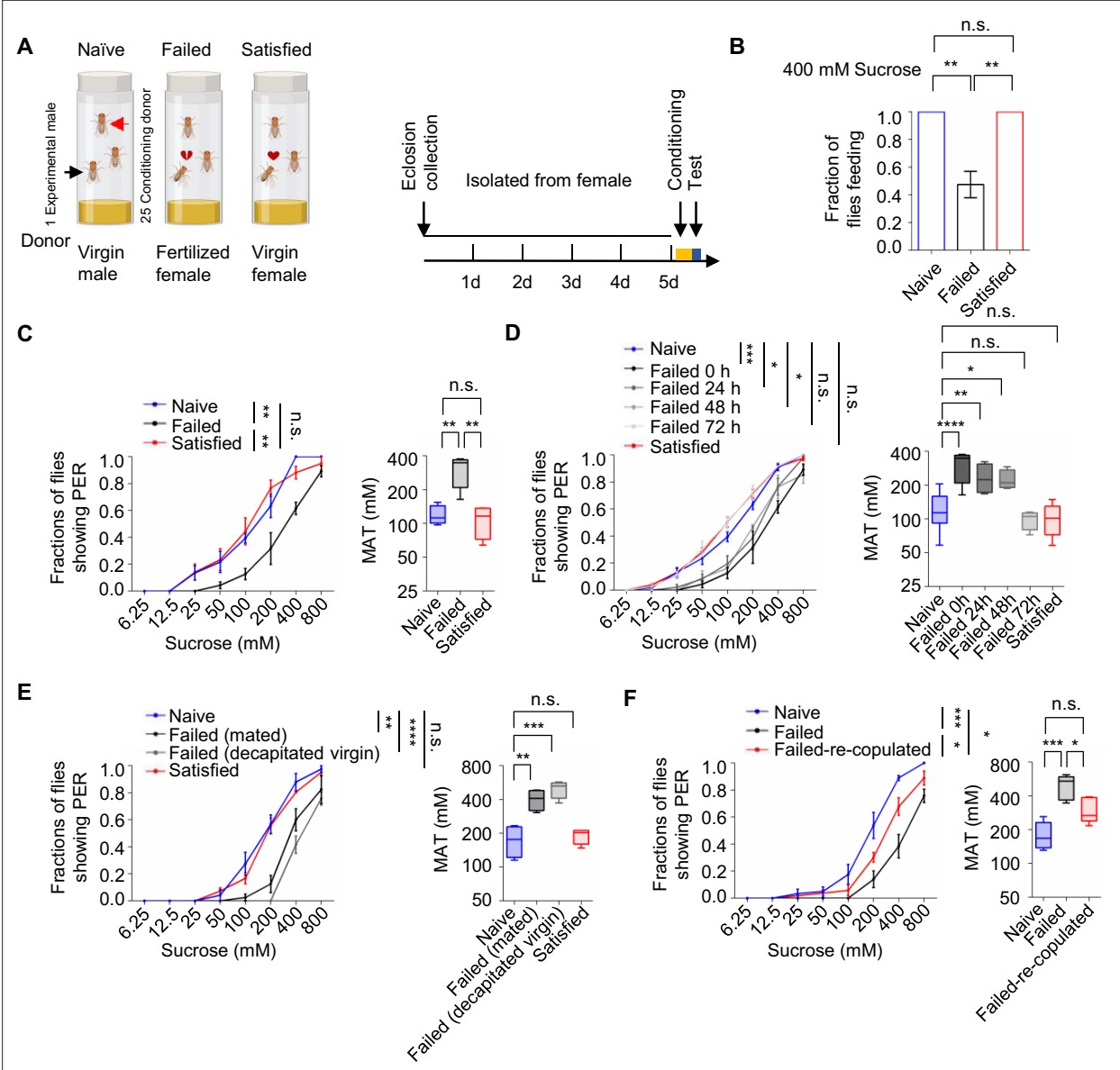

**Figure 1.** Sexual failure decreased sweet sensitivity. (**A**) Schematic illustrating courtship conditioning strategy. (**B**) Percentage of males that performed feeding to 400 mM sucrose (n=40). (**C**) Fraction of Naïve, Failed, and Satisfied male flies showing proboscis extension reflex (PER) responses to different concentrations of sucrose. The PER experiment was performed immediately after conditioning unless otherwise indicated. Left: average PER responses of Naïve (blue), Failed (black), and Satisfied (red) flies (n=40–48). Right: MAT (mean acceptance threshold; the sucrose concentration where 50% of the flies show PER), plotted as a function of sexual state. (**D**) Fraction of male flies showing PER responses to different concentrations of sucrose. The PER experiment was performed immediately, 24 hr, 48 hr, and 72 hr after conditioning (n=46–135). Naïve and Satisfied showed pooled data from 0 to 72 hr post-treatment (**E**) Fraction of male flies showing PER responses to different concentrations of sucrose (n=40–48). Failed male flies were generated by two methods: one group was subjected to mated females (with cis-vaccenyl acetate, cVA), and the other was subjected to decapitated virgin females (without cVA). (**F**) Fraction of male flies showing PER responses to different concentrations of sucrose (n=55–60). Failed-re-copulated males were generated by placing individual failed males (within 30 min post-treatment) with 25 virgin females in a food vial for 4.5 hr. Data are shown as means ± SEM. Error bars represent SEM. n.s., p>0.05; *p<0.05; **p<0.01; ***p<0.001, ****p<0.0001. One-way ANOVA (**B–F** MAT data) and two-way ANOVA (**C–F** PER data) were applied for statistical analysis.

The online version of this article includes the following source data and figure supplement(s) for figure 1:

**Source data 1.** Source data for *Figure 1*.

**Figure supplement 1.** Sexual failure decreased sweet sensitivity.

**Figure supplement 1—source data 1.** Source data for *Figure 1—figure supplement 1*.

developed previously (*Qi et al., 2015*). Despite the fact that the amount of sucrose consumed by individual flies was similar across different groups (*Figure 1—figure supplement 1A*), the proportion of males ever consuming any sucrose was significantly lower in the Failed group (*Figure 1B*), suggesting a reduced likelihood of meal initiation among these males. Notably, mating failure also compromised the flies' ability to withstand starvation, as Failed males died significantly faster under food deprivation (*Figure 1—figure supplement 1B*).

Since taste perception is closely linked to the initiation of feeding behavior, we next examined whether sexual experience affected sweet sensitivity using the PER assay. This well-established method quantified the tendency to initiate feeding behavior towards different gustatory stimuli (*Scott, 2005*; *Scott, 2018*). While three groups of male flies all exhibited dose-dependent responses to varying sucrose concentrations, males in the Failed group demonstrated a rightward shift of the concentration-response curve, indicating significantly reduced sweet sensitivity (*Figure 1C*, *Figure 1—figure supplement 1C*). Consistently, the Mean Acceptance Threshold (MAT), the sucrose concentration at which 50% of the population displayed a PER response, was significantly elevated in the Failed group (*Figure 1C*, *Figure 1—figure supplement 1C* right). Furthermore, we found that diminished sweet sensitivity extended to other sweet compounds such as fructose and glucose (*Figure 1—figure supplement 1D, E*), indicating a general reduction in sweet sensitivity and an overall elevation in the threshold to initiate feeding upon appetitive cues.

To determine the time course of this effect, we monitored PER responses in male flies at 24, 48, and 72 hr post-treatment. Sweet sensitivity remained suppressed for up to 48 hr in Failed males but returned to baseline levels by 72 hr (*Figure 1D*). To explore the mechanisms underlying this lasting change, we examined whether it could be attributed to the exposure to cis-vaccenyl acetate (cVA), a pheromone present on the cuticle of mated females but absent from virgin females (*Ejima et al., 2007*). Conditioning males with either mated females or decapitated virgin females produced similar reductions in sweet sensitivity (*Figure 1E*), suggesting that cVA is not essential for this behavioral effect. We then tested whether the reduced sweet sensitivity was a result of failed copulation. Failed males were then divided into two groups for additional treatments: one group remained untreated, while the other was allowed to copulate with virgin females. We found that the reduction in sweet sensitivity caused by sexual failure could be reversed through subsequent successful copulation. Such males showed significantly increased sugar sensitivity compared to those that did not have a chance to copulate at all (*Figure 1F*). In contrast, when previously mated flies experienced courtship failure, their sweet sensitivity was also significantly decreased, indicating that prior successful mating did not prevent the alteration in sweet sensitivity induced by subsequent courtship failure (*Figure 1—figure supplement 1F*). Overall, our findings reveal that sexual failure exerted a long-lasting yet reversible effect to reduce sweet sensitivity in male flies.

## Dopamine signaling mediated the suppression of sweet sensitivity following sexual failure

The long-lasting and reversible effect of sexual failure hinted at certain changes in flies' internal motivational state. Dopamine was reported to regulate PER behavior in a metabolic state-dependent manner (*Inagaki et al., 2012*; *Marella et al., 2012*). We thus hypothesized that dopamine might also play a key role in the taste plasticity triggered by sexual failure. To test this, we first used a pharmacological approach to reduce dopamine signaling in male flies by feeding them 3-iodotyrosine (3IY), a tyrosine hydroxylase-specific inhibitor that blocked dopamine synthesis (*Inagaki et al., 2012*), and then measured their PER responses. Flies treated with 3IY showed comparable sweet responses across all three experimental groups, with an overall reduction in sweet sensitivity compared to the control group, effectively abolishing the changes induced by sexual failure (*Figure 2A*). To further confirm the necessity of dopamine in modulating sexual experience-dependent sweet sensitivity, we employed a genetic approach by using $Th^{-/-}$ mutant flies, in which tyrosine hydroxylase, the rate-limiting enzyme in dopamine synthesis, was eliminated (*Deng et al., 2019*; *Friggi-Grelin et al., 2003*; *Marella et al., 2012*). As expected, $Th^{-/-}$ mutant flies in the Failed group exhibited sweet sensitivity indistinguishable from those in the Naïve and Satisfied groups (*Figure 2—figure supplement 1A*).

Next, we investigated whether elevating dopamine levels could restore diminished sweet sensitivity in males from the Failed group. We first tested the effect of exogenous dopamine on sweet sensitivity by injecting dopamine into the base of the flies' proboscis. Consistent with previous findings (*Inagaki*

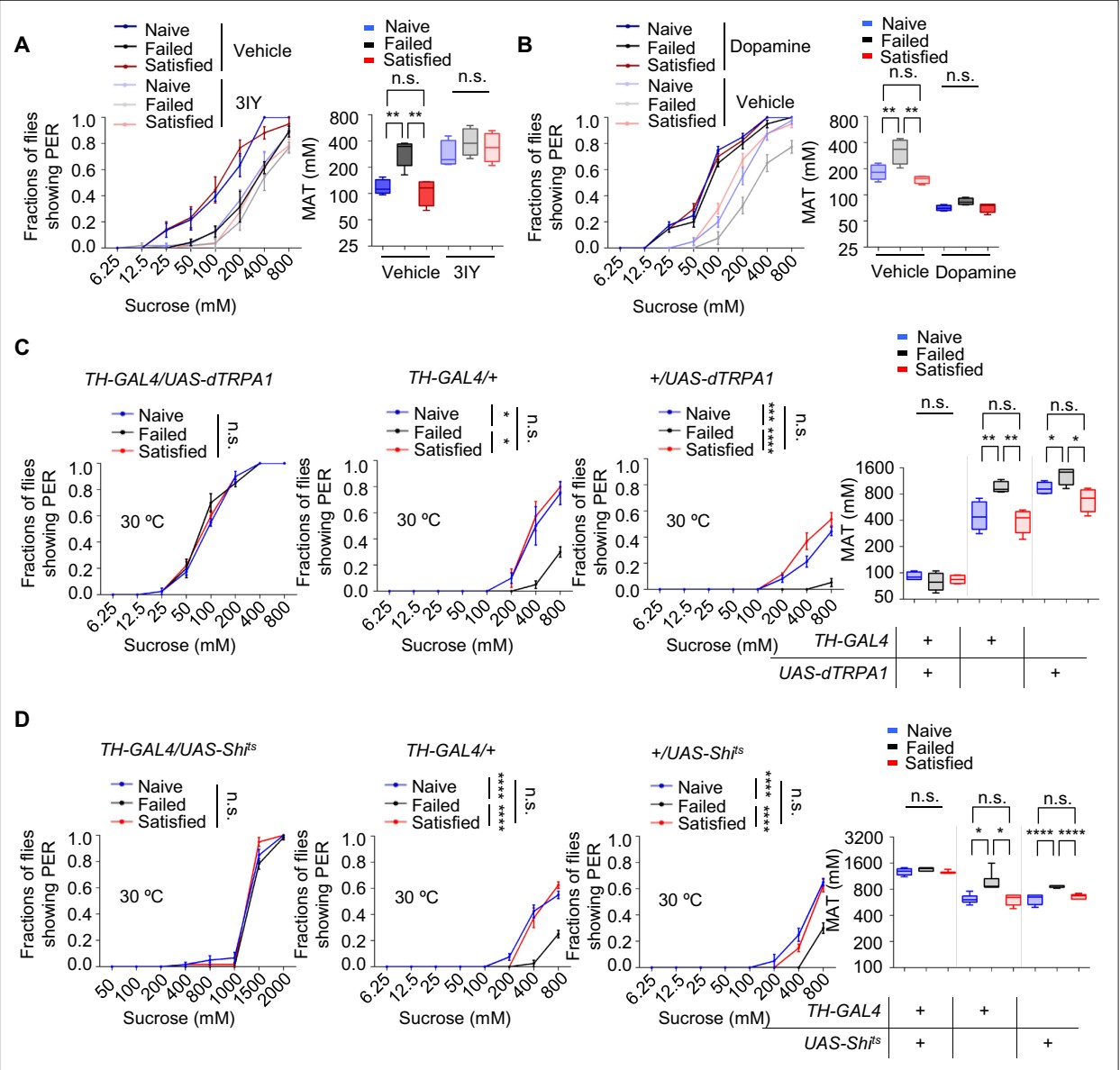

**Figure 2.** Dopamine signaling modulated the effect of sexual experience on sweet sensation. (**A**) Proboscis extension reflex (PER) responses in male flies fed with 3IY, compared to male flies fed with vehicle (n=40–48). (**B**) PER responses in male flies with dopamine injection, compared to male flies injected with vehicle (n=40). (**C, D**) PER responses in male flies with indicated genotypes at 30°C (n=48–60). Data are shown as means ± SEM. Error bars represent SEM. n.s., p>0.05; *p<0.05; **p<0.01; ***p<0.001; ****p<0.0001. Two-way ANOVA (**A–D** PER data) and one-way ANOVA (**A–D** MAT data) were applied for statistical analysis.

The online version of this article includes the following source data and figure supplement(s) for figure 2:

**Source data 1.** Source data for *Figure 2*.

**Figure supplement 1.** Dopamine but not serotonin signaling modulated sexual experience-dependent sweet sensitivity.

**Figure supplement 1—source data 1.** Source data for *Figure 2—figure supplement 1*.

*et al., 2012*; *Marella et al., 2012*), increasing dopamine levels enhanced sweet sensitivity (*Figure 2— figure supplement 1B*). We then measured PER behavior in Naïve, Failed, and Satisfied flies injected with either dopamine or ddH$_2$O (control). While ddH$_2$O-treated males from the Failed group displayed reduced sweet sensitivity compared to Naïve and Satisfied males, dopamine-treated Failed flies did not show such a reduction (*Figure 2B*), further suggesting that dopamine signaling is necessary for sexual failure to suppress sweet sensitivity.

Besides dopamine, serotonin, another important biogenic amine, has been reported to modulate motivational states and influence feeding behavior (*Owald and Waddell, 2015*; *Ries et al., 2017*; *Srinivasan et al., 2008*). Therefore, we examined whether serotonin also played a role in the sexual experience-dependent changes in sweet sensitivity. Feeding male flies with DL-p-chlorophenylalanine (pCPA) or methysergide, both blocking serotonin signaling (*Engelman et al., 1967*; *McCorvy et al., 2018*), did not affect the reduction in sweet sensitivity upon sexual failure (*Figure 2—figure supplement 1C, D*).

To further validate the involvement of dopamine signaling in sexual failure-induced behavioral changes, we used *TH-Gal4* to drive *UAS-dTRPA1* or *UAS-Shibire$^{ts}$* (*Shi$^{ts}$*) to transiently activate or inhibit dopaminergic neurons, respectively (*Deng et al., 2019*). Activation of dopaminergic neurons through the temperature-sensitive cation channel dTRPA1 (*Pulver et al., 2009*) increased sugar sensitivity in all male flies and eliminated differences among groups with different sexual experiences. Meanwhile, control genotypes expressing only *TH-Gal4* or *UAS-dTRPA1* still showed significantly reduced sugar sensitivity in the Failed group (*Figure 2C*). Additionally, males from the Failed group showed decreased sweet sensitivity across all genotypes at the permissive temperature (20°C) (*Figure 2—figure supplement 1E*). In contrast, blocking dopaminergic neurons via the temperature-sensitive dominant-negative Dynamin *Shibire$^{ts}$* (*Shi$^{ts}$*) (*Kitamoto, 2001*) decreased sugar sensitivity across all three groups and also completely suppressed the differences among groups (*Figure 2D*). In comparison, in the two control genotypes and under permissive temperature (20°C), sexual failure could still suppress sweet sensitivity (*Figure 2D*, *Figure 2—figure supplement 1F*). These findings demonstrate that reduced sweet sensitivity in sexually failed males is mediated by dopaminergic neurons.

## Dopamine signaling directly modulated the activity of Gr5a$^+$ gustatory neurons upon sexual failure

We next asked how dopamine signaling modulated taste sensitivity. Gustatory receptor neurons (GRNs) respond to specific taste stimuli, with Gr5a$^+$ neurons, which express sweet taste receptors, playing a key role in sweet perception (*Dahanukar et al., 2007*; *Thorne et al., 2004*; *Wang et al., 2004*). We hypothesized that behavioral changes following sexual failure might be linked to altered activities of sweet-sensing GRNs. To test this, we expressed calcium indicator GCaMP6m (*Chen et al., 2013*) in Gr5a$^+$ neurons and conducted in vivo calcium imaging. Gr5a$^+$ neurons project to SEZ in the fly brain, where we recorded calcium signals in response to sucrose stimulation. Sexual failure significantly reduced the responsiveness of Gr5a$^+$ neurons to sucrose, consistent with our behavioral observations (*Figure 3A–D*).

Given that dopamine signaling has been shown to modulate sweet sensitivity (*Inagaki et al., 2012*; *Marella et al., 2012*), it was plausible that dopamine signaling directly modulated the activity of Gr5a$^+$ neurons upon sexual failure. We thus examined projection patterns of dopaminergic neurons relative to those of Gr5a$^+$ neurons. Using *TH-Gal4* and *Gr5a-LexA*, we expressed *UAS-mCD8GFP* and *LexAop-rCD2RFP* in the same fly brains, respectively. The brain imaging data revealed that dopaminergic projections were closely aligned with Gr5a$^+$ GRNs in the SEZ (*Figure 3—figure supplement 1A*), suggesting possible synaptic connections of Gr5a$^+$ neurons and dopaminergic neurons.

We confirmed this idea using enhanced trans-synaptic reconstruction (nSyb-GRASP) (*Feinberg et al., 2008*; *Gordon and Scott, 2009*; *Macpherson et al., 2015*), which could visualize direct and active synaptic innervations between two groups of neurons. The nSyb-GRASP system relies on the expression of two split-GFP fragments on membranes of two neuronal populations, which fluoresce only upon the reassembly of these two fragments at active synapses. By employing the nSyb-GRASP system in Gr5a$^+$ neurons and dopaminergic neurons, we observed GFP signals in the SEZ (*Figure 3—figure supplement 1B*). Furthermore, in males that experienced mating failure, synaptic connections between dopaminergic neurons and Gr5a$^+$ GRNs were significantly weakened (*Figure 3E–F*). These results suggest that dopaminergic neurons directly innervate Gr5a$^+$ neurons and modulate their activity upon sexual failure.

This idea was further assessed by other complementary approaches. We expressed the photosensitive protein CsChrimson (*Klapoetke et al., 2014*) in dopaminergic neurons and GCaMP6m in Gr5a$^+$ neurons. Light activation of dopaminergic neurons significantly enhanced the calcium responses of Gr5a$^+$ neurons to sugar stimuli, both in Naïve and Failed groups (*Figure 3—figure supplement 1C, D*). Furthermore, by using a CaLexA reporter system (*Masuyama et al., 2012*), we assessed the

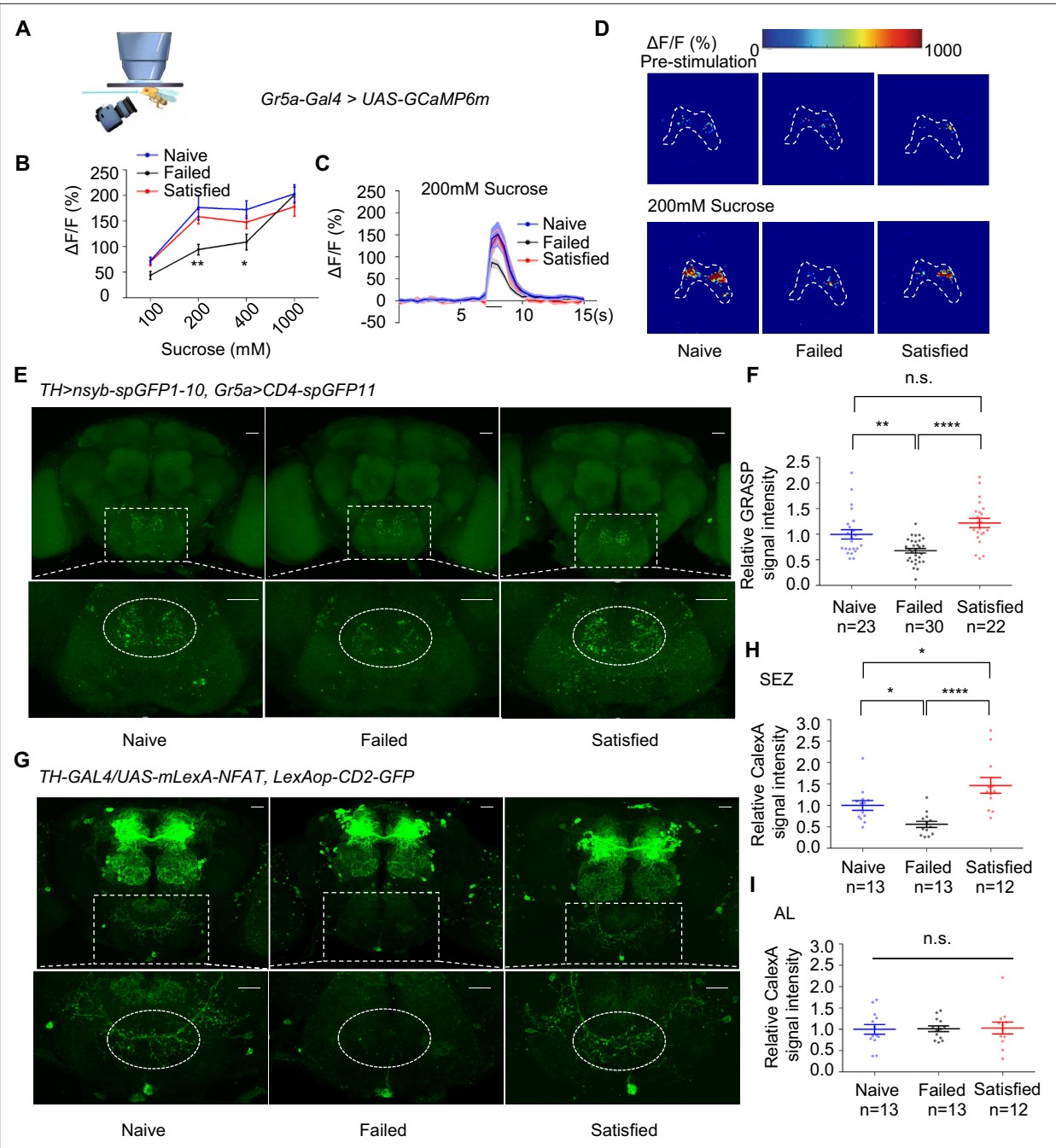

**Figure 3.** Dopamine neurons modulated the activity of Gr5a⁺ neurons upon sexual failure. (**A**) Schematic diagram of in vivo calcium imaging paradigm. (**B**) Quantification of the in vivo calcium responses of Gr5a⁺ neurons to the indicated concentrations of sucrose (n=5-18). (**C**) Averaged traces of the calcium responses to 200 mM sucrose. The horizontal black bar represents sucrose application. The solid lines represent the average trace, and shaded regions represent ± SEM. (**D**) Representative pseudocolor images of the calcium responses to 200 mM sucrose. The dashed area was quantified to measure the response. (**E**) Top: Representative images from the GRASP experiment between dopaminergic neurons and sweet Gustatory receptor neurons (GRNs). Bottom: Enlarged images of the subesophageal zone (SEZ) region seen on the top. Scale bars, 20 µm. (**F**) Average fluorescence reported by *TH >nsyb-spGFP1-10, Gr5a>CD4-spGFP11* (normalized to Naive) in the SEZ. Regions of interest (ROIs) are circled by dashed lines in (**E**). Lines represent mean ± SEM, and dots indicate values for individual flies. (**G**) Top: Representative images of CaLexA-induced GFP reporter in dopaminergic neurons. Bottom: Enlarged images of the SEZ region were shown on top. Scale bars, 20 µm. (**H**, **I**) Average fluorescence reported by *TH-GAL4/UAS-mLexA-NFAT, LexAop-CD2-GFP* (normalized to Naive) in the SEZ (**H**), in the AL (**I**). Regions of interest (ROIs) are circled by dashed lines in (**G**). Lines represent mean ± SEM, and dots indicate values for individual flies. Data are shown as means ± SEM. Error bars represent SEM. n.s., p>0.05; *p<0.05; **p<0.01; ***p<0.001, ****p<0.0001. Two-way ANOVA (**B**) and one-way ANOVA (**F**, **H**, **I**) were applied for statistical analysis.

*Figure 3 continued on next page*

*Figure 3 continued*

The online version of this article includes the following source data and figure supplement(s) for figure 3:

**Source data 1.** Source data for *Figure 3*.

**Figure supplement 1.** Dopaminergic neurons had functional connections with sweet gustatory receptor neurons (GRNs).

**Figure supplement 1—source data 1.** Source data for *Figure 3—figure supplement 1*.

impact of sexual experience on neural activity in male fruit flies. The results showed that sexual failure led to a significant reduction in CaLexA-mediated GFP signals in dopaminergic neurons in the SEZ but not in the antennal lobes (AL) (*Figure 3G–I*). In summary, these findings indicate that mating experience modulates sweet taste perception in flies by inhibiting the activity of specific dopamine neurons in the SEZ and reducing synaptic transmission between dopaminergic neurons and sweet-sensing Gr5a$^+$ GRNs.

## Two dopamine receptors, Dop1R1 and Dop2R, mediated the effect of sexual failure

We next explored the molecular mechanisms involved in the reduction of sweet sensitivity upon sexual failure. Four dopamine receptors (Dop1R1, Dop1R2, Dop2R, and DopEcR) have been identified in *Drosophila* (*Deng et al., 2019*; *Feng et al., 1996*; *Hearn et al., 2002*; *Srivastava et al., 2005*; *Sugamori et al., 1995*). Through screening of these receptors, we found that mutations in *Dop1R1* or *Dop2R* abolished the effect of sexual failure on sweet sensitivity, while mutations in *Dop1R2* or *DopEcR* did not prevent the decline in sweet sensitivity following sexual failure (*Figure 4A–F*). Intriguingly, although Dop2R and DopEcR are both required for starvation-driven enhancement of sugar sensitivity (*Inagaki et al., 2012*; *Marella et al., 2012*), Dop2R and Dop1R1 (but not DopEcR) were necessary for the suppression of sweet sensitivity after courtship failure. This dissociation implies that reproductive experience and metabolic state engage distinct dopaminergic circuits, possibly via different neuronal subsets, dopamine release dynamics, or receptor recruitment, to modulate peripheral sensitivity.

To further confirm the roles of Dop1R1 and Dop2R, we used RNA interference (RNAi) to knock down these receptors in flies' nervous system. In line with the above results, knockdown of Dop1R1 or Dop2R blocked the influence of sexual experience on sweet sensitivity, while knockdown of Dop1R2 or DopEcR had no significant effect (*Figure 4—figure supplement 1A–D*). These results suggest that reductions in sweet sensitivity induced by sexual failure are mediated by dopamine signaling acting on Dop1R1 and Dop2R receptors.

Next, we activated the neurons expressing Dop1R1 or Dop2R by expressing the bacterial sodium channel NaChBac (*Nitabach et al., 2006*). Constitutive activation via NaChBac not only prevented courtship failure-induced decline in sucrose responsiveness but also enhanced PER in Naïve flies compared to the GAL4/+controls (*Figure 5A, B*). Meanwhile, activating Dop1R2$^+$ or DopEcR$^+$ neurons did exhibit such an effect (*Figure 5—figure supplement 1A–C*).

We also tried to silence Dop1R1$^+$ or Dop2R$^+$ neurons using the hyperpolarizing potassium channel (Kir2.1) (*Baines et al., 2001*). Kir2.1-mediated inhibition suppressed sweet sensitivity across Naïve, Satisfied, and Failed groups and abolished any additional reduction in the Failed cohort (*Figure 5C, D*). These findings demonstrate that dopaminergic input through Dop1R1/Dop2R acts as a positive drive to Gr5a$^+$ sweet-sensing neurons and that sexual failure suppresses sweet sensitivity by attenuating this dopaminergic tone.

## Gr5a$^+$ neurons received dopaminergic modulation through Dop1R1 and Dop2R

The above discoveries raised a possibility that sweet-sensing Gr5a$^+$ neurons might receive direct dopaminergic modulation via Dop1R1 and Dop2R receptors, and that such a mechanism might be critical for the modulation of sweet sensitivity upon sexual failure. To investigate this possibility, we first examined the expression of Dop1R1 and Dop2R in Gr5a$^+$ neurons. Immunostaining confirmed that both receptors were expressed in the labellum (*Figure 6—figure supplement 1A*). Moreover, co-labeling experiments revealed a significant overlap of Dop1R1/Dop2R expression in Gr5a$^+$ neurons (*Figure 6A, B*).

Next, we assessed whether the expression of Dop1R1 and Dop2R in Gr5a+ neurons was functionally important for the reduction of sweet sensitivity upon sexual failure. By using RNAi to reduce Dop1R1 or Dop2R expression in Gr5a+ neurons, we found that males experiencing sexual failure no longer exhibited a reduction in sweet sensitivity compared to Naïve or successfully mated males (*Figure 6C, D*). In contrast, reducing the expression of another dopamine receptor, DopEcR, which was also partially expressed in Gr5a+ neurons (*Figure 6—figure supplement 1B*), did not prevent the reduction of sweet sensitivity upon sexual failure (*Figure 6—figure supplement 1C*). These data suggest that Dop1R1 and Dop2R in Gr5a+ neurons are indeed essential for the observed behavioral changes.

We then investigated whether Dop1R1 and Dop2R were required for the modulation of neuronal activity of Gr5a+ neurons. Following sexual failure, the calcium responses of Gr5a+ neurons to sugar stimuli were suppressed (*Figure 3C*). But this effect was eliminated when Dop1R1 or Dop2R expression was knocked down in Gr5a+ neurons (*Figure 6E*). In summary, our results indicate that Dop1R1 and Dop2R in sweet-sensing Gr5a+ neurons mediate the sexual experience-dependent regulation of sweet sensation by modulating the activity of these neurons.

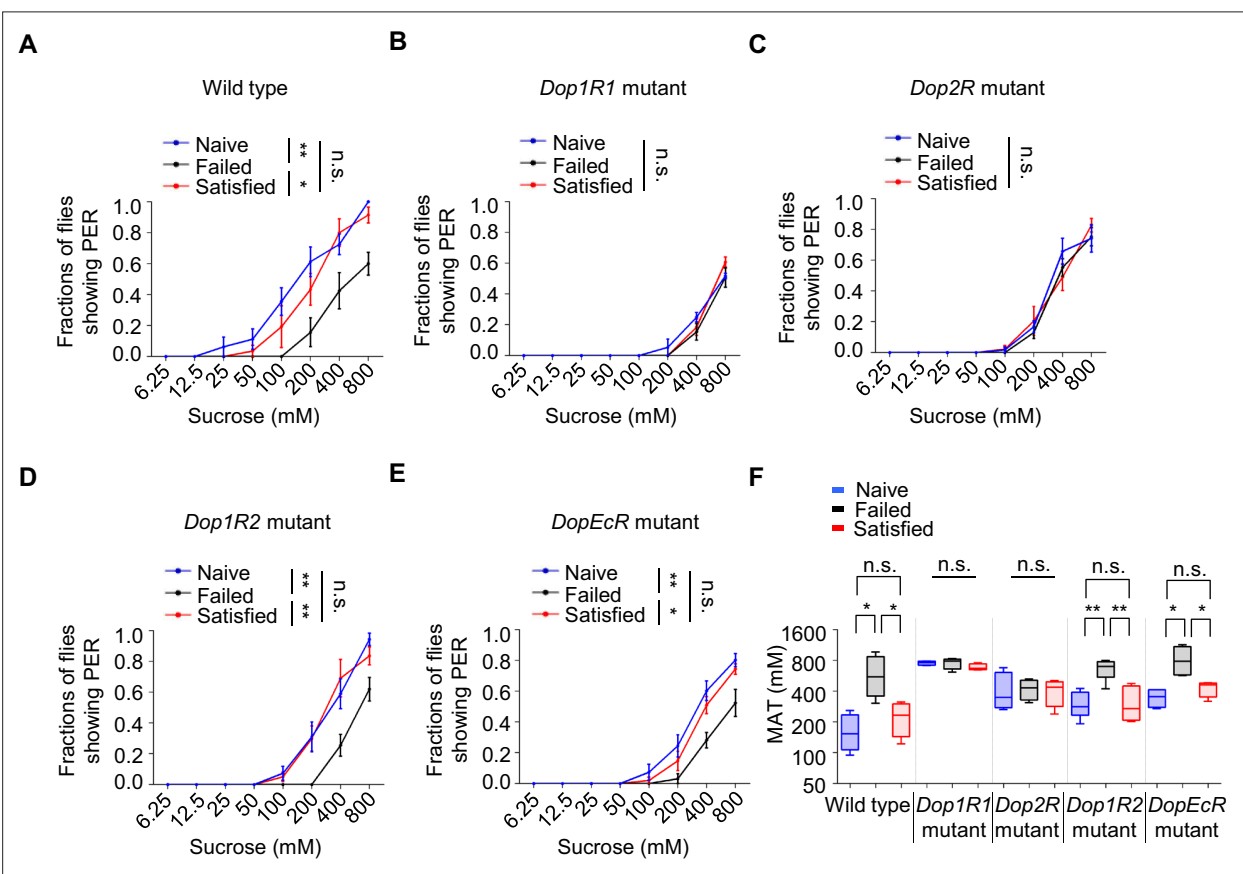

**Figure 4.** The effect of sexual failure was modulated through two dopamine receptors, Dop1R1 and Dop2R. (**A–E**) Proboscis extension reflex (PER) responses in male flies with indicated genotypes (n=40–60). (**F**) Mean acceptance Threshold threshold (MAT) calculation for the PER data from (**A–E**), plotted as a function of sexual state and type of flies. Data are shown as means ± SEM. Error bars represent SEM. n.s., p>0.05; *p<0.05; **p<0.01; ***p<0.001, ****p<0.0001. Two-way ANOVA (**A–E**) and one-way ANOVA (**F**) were applied for statistical analysis.

The online version of this article includes the following source data and figure supplement(s) for figure 4:

Source data 1. Source data for *Figure 4*.

Figure supplement 1. Dop1R1 and Dop2R were necessary for sexual failure-induced change in sweet sensitivity.

Figure supplement 1—source data 1. Source data for *Figure 4—figure supplement 1*.

## Discussion

Various innate behaviors are crucial for the survival and reproduction of animals, such as feeding, mating, aggression, sleep, and running away from predators (*Amir et al., 2015*; *Decoster et al., 2024*; *Hall, 1994*; *Hashikawa et al., 2016*; *Hergarden et al., 2012*; *Nair et al., 2023*; *Weber and Dan, 2016*; *Yapici et al., 2016*). However, when animals are exposed to competing and even conflicting needs of these innate behaviors, they have to adjust their behavioral output to balance these needs in a timely and precise manner. In the present study, we explored the potential link between two innate behaviors, mating and feeding. We found that failed experiences in sexual interactions suppressed sweet taste perception and, hence, feeding behavior in male fruit flies (*Figure 7*). The evolutionary significance of this behavioral plasticity remains unclear. It may represent an adaptive response where reproductive failure shifts time allocation away from feeding to conserve resources or seek alternative mating opportunities. Alternatively, it may indicate a maladaptive state of disrupted reward processing. Previous research has demonstrated that mating failure diminishes animals' motivational state, leading to widespread behavioral changes (*Beckwith et al., 2017*; *Jung et al., 2020*; *Ryvkin et al., 2024*; *Shen et al., 2023*; *Shohat-Ophir et al., 2012*). Our study extended this knowledge by showing that mating failure could also affect reward-related feeding behavior. This suggests that different instinctive behaviors might share a common reward mechanism, which exerts broader effects to influence animals' behavioral choices and physiological states.

Feeding provides energy critical for animals' survival, and it is not surprising that feeding is under tight and complex regulations (*Atasoy et al., 2012*; *Morton et al., 2006*; *Xu et al., 2008*; *Yapici et al., 2016*; *Zhan et al., 2016*). While a high volume of studies focus on mechanisms of how metabolic states modulate feeding behavior (*Ganguly et al., 2021*; *Inagaki et al., 2012*; *Marella et al., 2012*; *Pool and Scott, 2014*; *Shiu et al., 2022*), our study suggests the motivational drive of feeding, especially that influenced by mating experience. Previous investigations have shown that males who failed in sexual behaviors increase their ethanol preference (*Shohat-Ophir et al., 2012*), have an accelerated aging process (*Gendron et al., 2014*), and exhibit a sexual experience-induced preference behavior (*Shen et al., 2023*). Here, we demonstrated that sexual failure suppressed feeding behavior, one of the most basic homeostatic systems critical for animal fitness. These results collectively suggest that there may be a common neural mechanism that underlies the crosslink between mating failure and other innate behaviors.

Notably, we found that this behavioral coordination persisted for an extended duration and gradually faded over time, yet could be quickly reversed by successful mating. This suggests that the regulatory mechanism may modulate the internal state of the animal. A likely mechanism for this is through the reward system, as both successful mating and feeding are known to activate the reward system (*Burke et al., 2012*; *Dai et al., 2022*; *Schultz, 2010*; *Stuber and Wise, 2016*; *Zer-Krispil et al., 2018*). To support this idea, our study demonstrated that dopamine, a key neurotransmitter associated with reward (*Peters et al., 2021*; *Wise, 2004*), plays an integral role in mediating this behavioral regulation. Dopamine plays a central role in regulating internal motivational states. It has been shown that factors like social interactions, mating, hunger, and vigilance influence behavioral output through dopamine signaling (*Dai et al., 2022*; *de Jong et al., 2019*; *Inagaki et al., 2012*; *Marella et al., 2012*; *Schultz, 2010*; *Watabe-Uchida and Uchida, 2018*). Dopamine has been implicated in coding the valence of mating and feeding experiences (*Burke et al., 2012*; *Dai et al., 2022*; *Stuber and Wise, 2016*; *Zhang et al., 2019*). Thus, it is possible that the dopamine system acts as a motivation/reward center, allocating limited energy toward different behaviors in an internal state-dependent manner. This raises a critical question: does dopamine also mediate interactions among various innate behaviors, such as sleep, aggression, courtship, fear, and feeding, especially when these behaviors are in conflict? Additionally, it remains to be determined whether a single group of dopamine neurons mediates the crosslink amongst these behaviors, or whether distinct neuronal populations regulate separate behavioral interactions, and the mechanisms by which sexual failure selectively impacts different dopaminergic subgroups warrant further investigation.

Disturbances in innate behaviors have widespread and lasting effects on animals' overall behavioral patterns and physiological states (*Goldstein et al., 2018*; *Grzelka et al., 2023*; *Jourjine et al., 2016*; *Sun et al., 2023*). This cross-behavioral influence is likely connected to the concept of primal emotions. Emotions are a fundamental mechanism for behavioral regulation across species, not exclusive to more complex animals. In *Drosophila*, previous studies have shown that the flies have emotion-like

states such as stress and fear (*Anderson and Adolphs, 2014*; *Gu et al., 2019*). Our research suggests that mating failure may act as a trigger for negative emotions, affecting other behaviors like feeding and reward responses. This finding provides a new perspective on how emotions influence behavioral adaptation through neural mechanisms and offers a model for future research on emotion and behavior.

In summary, this study reveals the neural mechanisms through which mating failure suppresses sweet taste perception in *Drosophila* and establishes a foundation for investigating the complex interactions between emotions and the regulation of innate behaviors. Future research will explore how the dopamine system modulates the output of various innate behaviors under different emotional states and whether such a regulatory mechanism is conserved across species.

# Materials and methods

## Key resources table

| Reagent type (species) or resource | Designation | Source or reference | Identifiers | Additional information |
|---|---|---|---|---|
| Genetic reagent (*D. melanogaster*) | TH-GAL4 | PMID:30799021 | | Gift from Yi Rao |
| Genetic reagent (*D. melanogaster*) | UAS-dTRPA1 | other | | Gift from David Anderson |
| Genetic reagent (*D. melanogaster*) | UAS-Shi^ts | other | | Gift from David Anderson |
| Genetic reagent (*D. melanogaster*) | Gr5a-GAL4 | Bloomington *Drosophila* Stock Center | Cat: #57592 | |
| Genetic reagent (*D. melanogaster*) | UAS-GCaMP6m | Bloomington *Drosophila* Stock Center | Cat: #42748 | |
| Genetic reagent (*D. melanogaster*) | Gr5a-LexA | Bloomington *Drosophila* Stock Center | Cat: #93014 | |
| Genetic reagent (*D. melanogaster*) | UAS-nSyb-spGFP1-10, lexAop-CD4-spGFP11 | Bloomington *Drosophila* Stock Center | Cat: #64314 | |
| Genetic reagent (*D. melanogaster*) | UAS-CalexA | Bloomington *Drosophila* Stock Center | Cat: #66542 | |
| Genetic reagent (*D. melanogaster*) | Dop1R1-/- mutant | PMID:30799021 | | Gift from Yi Rao |
| Genetic reagent (*D. melanogaster*) | Dop2R-/- mutant | PMID:30799021 | | Gift from Yi Rao |
| Genetic reagent (*D. melanogaster*) | Dop1R2-/- mutant | PMID:30799021 | | Gift from Yi Rao |
| Genetic reagent (*D. melanogaster*) | DopEcR-/- mutant | PMID:30799021 | | Gift from Yi Rao |
| Genetic reagent (*D. melanogaster*) | Dop1R1-GAL4 | PMID:30799021 | | Gift from Yi Rao |
| Genetic reagent (*D. melanogaster*) | Dop2R-GAL4 | PMID:30799021 | | Gift from Yi Rao |
| Genetic reagent (*D. melanogaster*) | UAS-NaChBac | Bloomington *Drosophila* Stock Center | Cat: #9469 | |
| Genetic reagent (*D. melanogaster*) | UAS-Kir2.1 | Bloomington *Drosophila* Stock Center | Cat: #6595 | |
| Genetic reagent (*D. melanogaster*) | 10×UAS-IVS-mCD8::RFP, 13×LexAop2-mCD8::GFP | Bloomington *Drosophila* Stock Center | Cat: #32229 | |
| Genetic reagent (*D. melanogaster*) | UAS-GCaMP6m | Bloomington *Drosophila* Stock Center | Cat: #42750 | |
| Genetic reagent (*D. melanogaster*) | UAS-Dop1R1 RNAi | Tsinghua Fly Center | Cat: #4278 | |
| Genetic reagent (*D. melanogaster*) | UAS-Dop2R RNAi | Tsinghua Fly Center | Cat: #2141 | |
| Genetic reagent (*D. melanogaster*) | Th-/- mutant | PMID:30799021 | | Gift from Yi Rao |
| Genetic reagent (*D. melanogaster*) | LexAop-rCD2RFP, UAS-mCD8GFP | Bloomington *Drosophila* Stock Center | Cat: #67093 | |
| Genetic reagent (*D. melanogaster*) | LexAop-GCaMP6m | other | | Gift from Yufeng Pan |
| Genetic reagent (*D. melanogaster*) | UAS-CsChrimson | other | | Gift from Yufeng Pan |
| Genetic reagent (*D. melanogaster*) | elav-GAL4 | Bloomington *Drosophila* Stock Center | Cat: #25750 | |

*Continued on next page*

*Continued*

| Reagent type (species) or resource | Designation | Source or reference | Identifiers | Additional information |
|---|---|---|---|---|
| Genetic reagent (*D. melanogaster*) | UAS-Dop1R2 RNAi | Tsinghua Fly Center | Cat: #5285 | |
| Genetic reagent (*D. melanogaster*) | UAS-DopEcR RNAi | Tsinghua Fly Center | Cat: #3275 | |
| Genetic reagent (*D. melanogaster*) | Dop1R2-GAL4 | PMID:30799021 | | Gift from Yi Rao |
| Genetic reagent (*D. melanogaster*) | DopEcR-GAL4 | PMID:30799021 | | Gift from Yi Rao |
| Genetic reagent (*D. melanogaster*) | UAS-mCD8::GFP | Bloomington *Drosophila* Stock Center | Cat: #5137 | |
| Antibody | anti-GFP (mouse) | Sigma-Aldrich | Cat: #G6539 | |
| Antibody | anti-GFP (mouse) | Abcam | Cat: #ab1218 | |
| Antibody | anti-DsRed (rabbit) | Takara Bio | Cat: #632496 | |
| Antibody | anti-GFP (rabbit) | Thermo Fisher Scientific | Cat: #A11122 | |
| Antibody | Goat anti-mouse Alexa Fluor 488 | Thermo Fisher Scientific | Cat: #A11029 | |
| Antibody | Goat anti-rabbit Alexa Fluor 546 | Thermo Fisher Scientific | Cat: #A11010 | |
| Antibody | Goat anti-rabbit Alexa Fluor 488 | Thermo Fisher Scientific | Cat: #A11034 | |
| Chemical compound | Sucrose | Sigma-Aldrich | Cat: #S0389 | |
| Chemical compound | 3-Iodo-L-tyrosine | Sigma-Aldrich | Cat: #I8250 | |
| Chemical compound | Dopamine | Aladdin | Cat: #A303863 | |
| Chemical compound | NaCl | Sigma-Aldrich | Cat: #S3014 | |
| Chemical compound | $MgCl_2$ | Sigma-Aldrich | Cat: #M4880 | |
| Chemical compound | $NaHCO_3$ | Sigma-Aldrich | Cat: #S5761 | |
| Chemical compound | $NaH_2PO_4$ | Sigma-Aldrich | Cat: #S3139 | |
| Chemical compound | KCl | Sigma-Aldrich | Cat: #P9541 | |
| Chemical compound | $CaCl_2$ | Sigma-Aldrich | Cat: #C5670 | |
| Chemical compound | HEPES | Sigma-Aldrich | Cat: #54457 | |
| Chemical compound | Paraformaldehyde | Electron Microscopy | Cat: #15713 | |
| Chemical compound | Calf serum | Thermo Fisher Scientific | Cat: #16010159 | |
| Chemical compound | Triton X-100 | Sigma-Aldrich | Cat: #T8787 | |
| Chemical compound | D-Fructose | Tokyo Chemical Industry | Cat: #F0060 | |
| Chemical compound | D-Glucose | Tokyo Chemical Industry | Cat: #G0048 | |
| Chemical compound | DL-p-chlorophenylalanine | Sigma-Aldrich | Cat: #c6506 | |
| Chemical compound | Methysergide maleate | Abcam | Cat: #ab120530 | |
| Chemical compound | All trans-retinal | Sigma-Aldrich | Cat: #R2500 | |
| Commercial assay or kit | Graduated glass capillary | VWR | Cat: #53432–604 | |
| Software, algorithm | Fiji/ImageJ | NIH | RRID:SCR_003070 | |
| Software, algorithm | GraphPad Prism 9 | GraphPad Software | RRID:SCR_002798 | |

## Flies

Flies were maintained in vials with standard cornmeal food at 25°C and 60% humidity in a 12 hr:12 hr light: dark cycle unless otherwise indicated. For temperature-sensitive flies (Shibire[ts] and dTRPA1), male flies were kept at 20°C before the behavioral tests. Fly strains used in the manuscript:

*elav-GAL4* (#25750), *UAS-mCD8::GFP* (#5137), *LexAop-rCD2RFP, UAS-mCD8GFP* (#67093), *10×UAS-IVS-mCD8::RFP, 13xLexAop2-mCD8::GFP* (#32229), *UAS-GCaMP6m* (#42748), *UAS-GCaMP6m* (#42750), *UAS-CalexA* (#66542), *UAS-NaChBac* (#9469), *UAS-Kir2.1* (#6595), *Gr5a-GAL4* (#57592), *Gr5a-LexA* (#93014), *UAS-nSyb-spGFP1-10, lexAop-CD4-spGFP11* (#64314) were obtained from the Bloomington *Drosophila* Stock Center at Indiana University. *UAS-Shi[ts]* and *UAS-dTRPA1* were

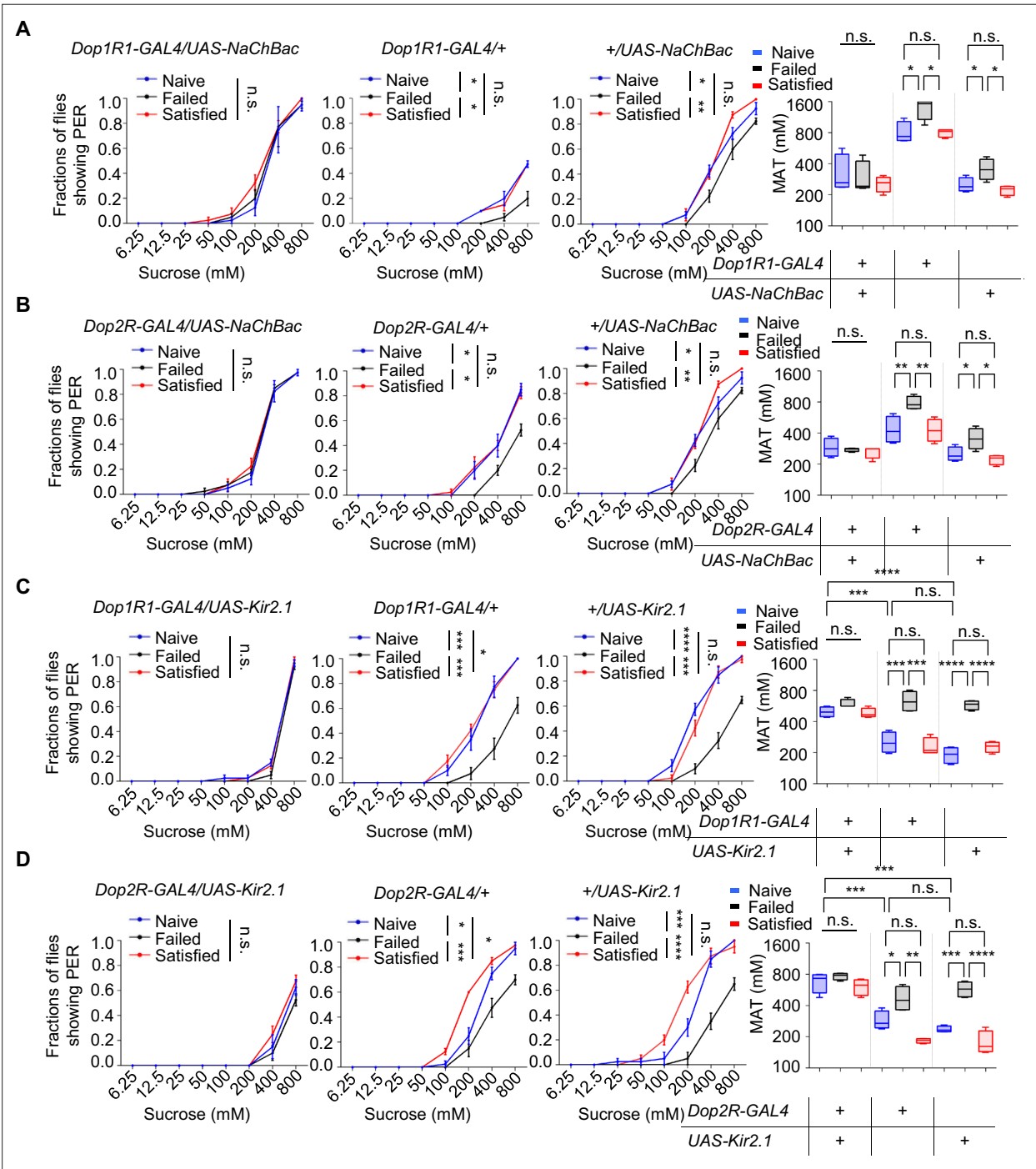

**Figure 5.** Dop1R1⁺ and Dop2R⁺ neurons regulated sexual experience-dependent sweet sensitivity. (**A**–**D**) Proboscis extension reflex (PER) responses in male flies with indicated genotypes (n=40). Data are shown as means ± SEM. Error bars represent SEM. n.s., p>0.05; *p<0.05; **p<0.01; ***p<0.001; ****p<0.0001. Two-way ANOVA (**A**–**D** PER data) and one-way ANOVA (**A**–**D** MAT data) were applied for statistical analysis.

The online version of this article includes the following source data and figure supplement(s) for figure 5:

Source data 1. Source data for *Figure 5*.

Figure supplement 1. Dop1R2⁺ or DopEcR⁺ neurons did not affect sweet sensitivity following sexual failure.

Figure supplement 1—source data 1. Source data for *Figure 5—figure supplement 1*.

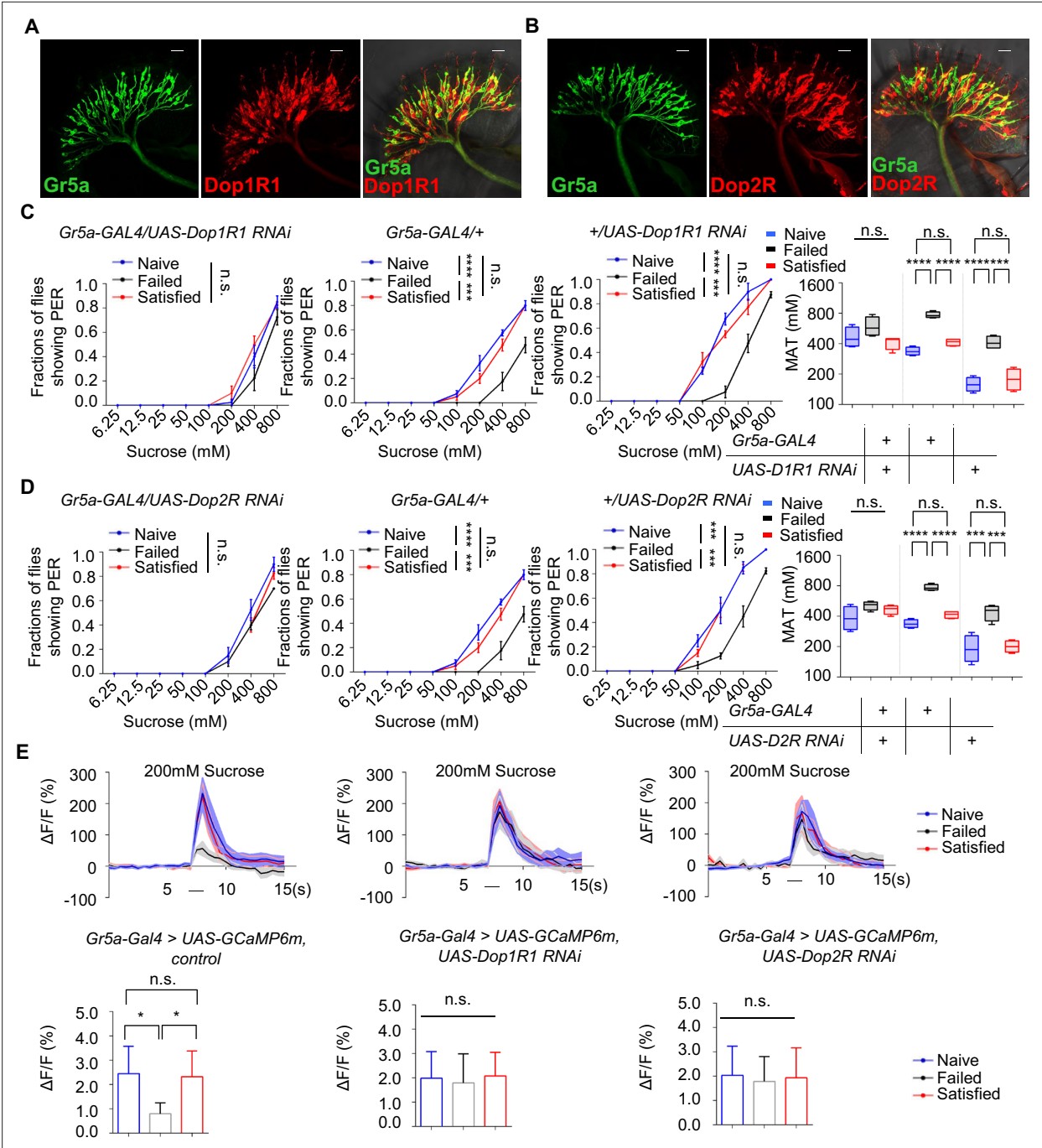

**Figure 6.** Gr5a⁺ neurons received dopaminergic modulation through Dop1R1 and Dop2R. (**A, B**) The expression of membrane-bound GFP (mCD8GFP, Green) in Gr5a⁺ neurons driven by *Gr5a-LexA* and membrane-bound RFP (mCD8RFP, Red) in dopamine receptor neurons driven by *Dop1R1-Gal4* (**A**), and *Dop2R-Gal4* (**B**). Scale bars, 20 μm. (**C, D**) PER responses in male flies with indicated genotypes (n=40). (**E**) Averaged traces of the calcium responses to 200 mM sucrose in male flies with indicated genotypes are shown on top. Quantification of the calcium responses was shown on the bottom (n=6–7). The horizontal black bars represent sucrose application. The solid lines represent the average trace, and shaded regions represent ± SEM. Data are shown as means ± SEM. Error bars represent SEM. n.s., p>0.05; *p<0.05; **p<0.01; ***p<0.001; ****p<0.0001. Two-way ANOVA (**C, D** PER data) and one-way ANOVA (**C, D** MAT data, **E**) were applied for statistical analysis.

The online version of this article includes the following source data and figure supplement(s) for figure 6:

**Source data 1.** Source data for *Figure 6*.

**Figure supplement 1.** Dopaminergic modulation of Gr5a⁺ neurons following sexual failure did not require DopEcR.

**Figure supplement 1—source data 1.** Source data for *Figure 6—figure supplement 1*.

kindly shared by David Anderson (California Institute of Technology). *Dop1R1^{-/-}* mutant, *Dop1R2^{-/-}* mutant, *Dop2R^{-/-}* mutant, *DopEcR^{-/-}* mutant, *Th^{-/-}* mutant, *Dop1R1-GAL4*, *Dop1R2-GAL4*, *Dop2R-GAL4*, *DopEcR-GAL4*, *TH-GAL4* were kindly shared by Yi Rao (Peking University). *LexAop-GCaMP6m* and *UAS-CsChrimson* were kindly shared by Yufeng Pan (Southeast University). *UAS-Dop1R1 RNAi* (#4278), *UAS-Dop1R2 RNAi* (#5285), *UAS-Dop2R RNAi* (#2141), and *UAS-DopEcR RNAi* (#3275) were obtained from the Tsinghua Fly Center.

## Conditioning protocols

We used a modified courtship conditioning protocol (*McBride et al., 1999*; *Shohat-Ophir et al., 2012*) to generate Naïve, Failed, and Satisfied males. The Failed group was generated by exposing virgin males to either mated females or decapitated virgin females. Individual virgin males were placed with either 25 mated females or 25 decapitated virgin females in a food vial for 4.5 hr. To generate mated females for the conditioning, 50 males were added to the 25 virgin females for 16 hr before the conditioning. The decapitated female flies were prepared freshly before assays. The Satisfied group was generated by exposing virgin males to virgin females. Individual virgin males were placed with 25 virgin females in a food vial for 4.5 hr. The Naïve group consisted of the age-matched virgin males kept isolated from females until the assays.

## The MAFE assay

The MAFE assay was carried out as described previously (*Qi et al., 2015*). Briefly, individual male flies were gently introduced and immobilized in a 200 µl pipette tip. The end of each pipette tip was shaped to expose the fly's proboscis. Flies were first sated with sterile water and then presented with 400 mM sucrose filled in a graduated glass capillary (VWR, #53432–604). The sucrose was presented until the flies no longer responded to a series of 10 sucrose stimuli. The ingested volume was calculated based on the volume change before and after the feeding.

## PER assay

PER was assayed as described previously (*Qi et al., 2015*). Individual male flies were mounted into a 200 µl pipette tip. Flies were first sated with sterile water and then subjected to different sugar solutions, each tested three times. Flies showing PER responses to at least 1 of 3 trials were considered positive to that sugar concentration. For drug treatment experiments, flies were maintained on food containing a drug for 2 days. Drugs used for treatment included 3-Iodo-L-tyrosine (3IY) (Sigma-Aldrich, 10 mg/ml), DL-p-chlorophenylalanine (pCPA) (Sigma-Aldrich, 10 mg/ml), and Methysergide maleate (Abcam, 25 mg/ml). For the dopamine injection experiment, Dopamine (Aladdin 0.1–10 ng/µl) was injected into the proboscis of each male fly using a glass micropipette. The standard PER assay was carried out at 25°C. For Shibire^{ts} and dTRPA1 experiments, flies were raised at 20°C, and assays were performed at either 20°C or 30°C. For PER assays performed at 24, 48, and 72 hr post-treatment, male flies were conditioned and then transferred into vials containing standard food before the behavioral assays. For Failed-re-copulated experiments, individual Failed male flies (within 30 min post-treatment) were allowed to mate with 25 virgin females in a food vial for 4.5 hr. For Satisfied-Failed experiments, individual Satisfied male flies (72 hr post-treatment) were exposed to 25 mated females for 4.5 hr. All PER assays had n >4 replicates with 10–15 flies per replicate unless otherwise stated.

## MAT calculation

The method to calculate MAT has been thoroughly described in previous papers (*Inagaki et al., 2012*; *Inagaki et al., 2014*). In general, sigmoid interpolation was performed to fit the results into sigmoidal curves. The sigmoid curves for sugar were defined as follows:

$$F_S = \frac{1}{1 + E\left(-\alpha_s log_2 \frac{s_{con}}{s_{50}}\right)}$$

$F_S$: Fraction of flies showing the PER on different sugar concentrations.
Scon: Sugar concentration.
$S_{50}$: Sucrose concentration where 50% of flies show PER

$\alpha_S$: Slope of the sigmoid curve

Based on the experimentally obtained results ($F_S$, Scon), $S_{50}$ and $\alpha_S$ were chosen to fit the results. For all experimental results, fitting based on nonlinear regression was calculated with Python.

### Starvation resistance assay

Starvation resistance of individual flies was monitored using DAMS (Trikinetics). Briefly, individual male flies were lightly anesthetized and introduced into 5×65 mm polycarbonate tubes (Trikinetics). One end of these tubes was filled with 2% agar or 5% sucrose, and the other end was blocked with cotton wool. These polycarbonate tubes were then inserted into DAMS monitors and kept in fly incubators during the course of the experiments.

## Immunohistochemistry

Fly brains or labella were dissected in PBS on ice and fixed in 4% paraformaldehyde at room temperature for 60 min. To examine GRASP signals, the fly brains were dissected 20 hr post-treatment. After rinsing with PBS for three times, the fixed brains or labella were blocked in Penetration/Blocking Buffer (10% Calf Serum and 0.5% Triton X-100 in PBS) at room temperature for 90 min and incubated with primary antibodies at 4°C for 24 hr. The samples were then rinsed with Washing Buffer (0.5% Triton X-100 in PBS) at room temperature for 3×30 min and incubated with secondary antibodies at 4°C for 24 hr. After another three rinses with Washing Buffer, the brains or labella were mounted in Fluoroshield (Sigma-Aldrich) for imaging. Images were acquired with Nikon C2 confocal microscope (×40/0.80 w DIC N2).

The antibodies were used as follows: rabbit anti-GFP (1:1000, Thermo Fisher Scientific), mouse anti-GFP (1: 1000, Abcam), mouse anti-GFP (1: 500, Sigma-Aldrich), rabbit anti-DsRed (1:1000, Takara Bio), Goat anti-rabbit Alexa Fluor 488 (1:1000, Thermo Fisher Scientific), Goat anti-mouse Alexa

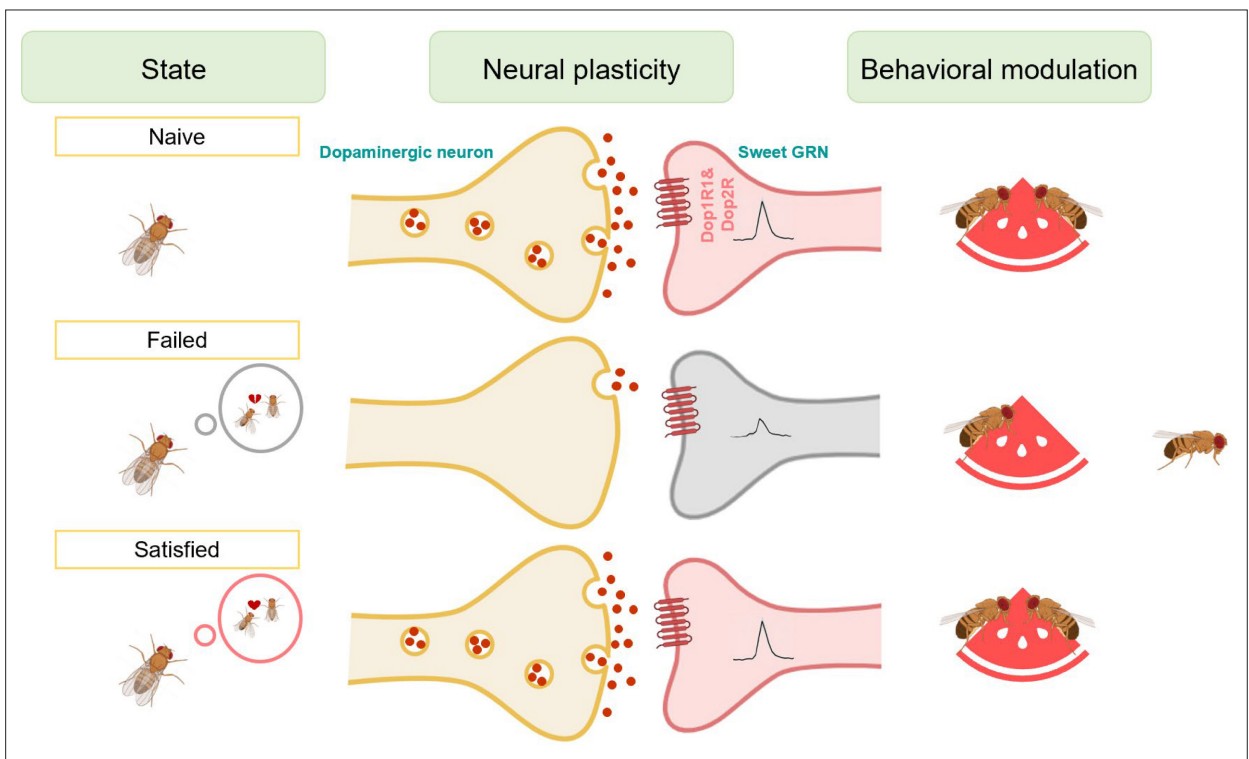

**Figure 7.** Working model for the sexual failure-dependent modulation of sweet perception. In the subesophageal zone (SEZ) of the fly brain, dopaminergic neurons form synaptic connections with sweet-sensing Gr5a⁺ neurons, which express the dopamine receptors Dop1R1 and Dop2R. This connectivity enables Gr5a⁺ neurons to receive dopaminergic modulation, enhancing the flies' sweet sensitivity. However, after mating failure, the activity of these SEZ dopaminergic neurons is selectively inhibited, and their synaptic connections with Gr5a⁺ neurons are weakened. This reduction in dopaminergic signaling ultimately decreases the flies' sweet taste perception. This mechanism highlights a specific pathway by which unsuccessful mating experiences influence sensory processing through neural plasticity in dopamine-associated circuits.

Fluor 488 (1:1000, Thermo Fisher Scientific), Goat anti-rabbit Alexa Fluor 546 (1:1000, Thermo Fisher Scientific).

## Calcium imaging

The in vivo calcium imaging experiment was performed as previously described (*Qi et al., 2021*). Flies were briefly anesthetized on ice and mounted onto transparent plastic tape. The cuticle part around the Gr5a$^+$ region of the brain was gently removed with forceps, and the brain was bathed in sugar-free adult hemolymph-like solution (AHL, 108 mM NaCl, 8.2 mM MgCl$_2$, 26 mM NaHCO$_3$, 1 mM NaH$_2$PO$_4$, 5 mM KCl, 2 mM CaCl$_2$, 5 mM HEPES, pH 7.3). A micro manipulator delivered sucrose to the proboscis of the fly, and the actual feeding bouts were imaged by a digital camera (FLIR USB 2.0, Point Gray). The calcium signals of Gr5a$^+$ neurons were collected by a Nikon C2 confocal microscope, with a water immersion objective lens (×40/0.80 w DIC N2).

For CsChrimson experiments, newly enclosed flies were maintained in vials with standard cornmeal food for 3 days and transferred to standard cornmeal food supplemented with 400 µM all-trans-retinal (Sigma-Aldrich) for 2 days before experiments. CsChrimson activation was provided by a fiber-coupled 625 nm LED (M625L3, Thorlabs). Image analyses were performed in ImageJ and plotted in GraphPad Prism 6 or MATLAB. The ratio changes were calculated using the following formula: $\Delta F/F = [F - F_0]/F_0$, where F was the peak intensity, and $F_0$ was the average intensity at baseline.

## Statistical analysis

Data presented in this study were all verified for normal distribution by the D'Agostino-Pearson omnibus test. In one-way ANOVA (for comparisons among three or more groups) tests, statistical significance was corrected using post hoc Tukey's test. The two-way ANOVA (for comparisons with more than one variant) tests use post hoc Bonferroni correction.

## Acknowledgements

We thank all Wang Lab members for helpful discussions and technical assistance. Ye Wu and Tingting Song provided scientific and administrative support in the laboratory. We thank the Bloomington *Drosophila* Stock Center at Indiana University and the Tsinghua Fly Center for fly stocks and reagents. This study was funded by the National Natural Science Foundation of China (32071006 for LW and No. 32271008 for RH), startup funds from Shenzhen Bay Laboratory, and the Chinese Institutes for Medical Research.

## Additional information

### Funding

| Funder | Grant reference number | Author |
| --- | --- | --- |
| National Natural Science Foundation of China | 32071006 | Liming Wang |
| National Natural Science Foundation of China | 32271008 | Rui Huang |

The funders had no role in study design, data collection and interpretation, or the decision to submit the work for publication.

### Author contributions

Gaohang Wang, Resources, Data curation, Formal analysis, Investigation, Methodology, Writing – original draft; Wei Qi, Investigation; Rui Huang, Resources, Formal analysis, Supervision, Funding acquisition, Investigation, Methodology, Writing – original draft, Project administration; Liming Wang, Conceptualization, Resources, Supervision, Funding acquisition, Writing – original draft, Project administration, Writing – review and editing

### Author ORCIDs

Liming Wang https://orcid.org/0000-0002-7256-8776

Reviewer #1 (Public review): https://doi.org/10.7554/eLife.105094.3.sa1
Reviewer #2 (Public review): https://doi.org/10.7554/eLife.105094.3.sa2
Reviewer #3 (Public review): https://doi.org/10.7554/eLife.105094.3.sa3
Author response https://doi.org/10.7554/eLife.105094.3.sa4

## Additional files

### Supplementary files
MDAR checklist

### Data availability
All data generated or analysed during this study are included in the manuscript and supporting files; source data files have been provided for *Figures 1–6* (including figure supplements) which contain the numerical data used to generate the figures.

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
