## [Editor Report · eLife Assessment]

This study provides **valuable** findings on the effects of mating experience on sweet taste perception. The data as presented provide **convincing** evidence that the dopaminergic signaling-mediated reward system underlies this mating state-dependent behavioral modulation. The work will interest neuroscientists and particularly biologists working on neuromodulation and the effects of internal states on sensory perception.

---

## [Referee Report · Reviewer #1 (Public review)]

Wang et al. investigated how sexual failure influences sweet taste perception in male *Drosophila*. The study revealed that courtship failure leads to decreased sweet sensitivity and feeding behavior via dopaminergic signaling. Specifically, the authors identified a group of dopaminergic neurons projecting to the subesophageal zone that interact with sweet-sensing Gr5a+ neurons. These dopaminergic neurons positively regulate the sweet sensitivity of Gr5a+ neurons via DopR1 and Dop2R receptors. Sexual failure diminishes the activity of these dopaminergic neurons, leading to reduced sweet taste sensitivity and sugar feeding behavior in the male flies. These findings highlight the role of dopaminergic neurons in integrating reproductive experiences to modulate appetitive sensory responses.

Previous studies have explored the dopaminergic-to-Gr5a+ neuronal pathways in regulating sugar feeding under hunger conditions. Starvation has been shown to increase dopamine release from a subset of TH-GAL4 labeled neurons, known as TH-VUM, in the subesophageal zone. This enhanced dopamine release activates dopamine receptors in Gr5a+ neurons, heightening their sensitivity to sugar and promoting sucrose acceptance in flies. Since the function of the dopaminergic-to-Gr5a+ circuit motif has been well established, the primary contribution of Wang et al. is to show that mating failure in male flies can also engage this circuit to modulate sugar feeding behavior. This contribution is valuable because it highlights the role of dopaminergic neurons in integrating diverse internal state signals to inform behavioral decisions.

An intriguing discrepancy between Wang et al. and earlier studies lies in the involvement of dopamine receptors in Gr5a+ neurons. Prior research has shown that Dop2R and DopEcR, but not DopR1, mediate starvation-induced enhancement of sugar sensitivity in Gr5a+ neurons. In contrast, Wang et al. report that DopR1 and Dop2R, but not DopEcR, are involved in the mating failure-induced suppression of sugar sensitivity in these neurons. Further investigation is needed to clarify how dopamine selectively engages different receptor types depending on internal state.

The data in this revised version are largely convincing and support the authors' conclusions. However, I remain concerned about the results shown in Figure 6E. The authors show that knocking down DopR1 or Dop2R in Gr5a+ neurons restores sucrose-evoked activity in Failed flies to levels seen in Naive and Satisfied animals. This appears to contradict the proposed model, in which these receptors positively modulate Gr5a+ activity through dopaminergic input. If dopamine signaling is reduced in Failed flies, further receptor knockdown should have no effect or further reduce activity-not restore it. I encourage the authors to clarify this apparent inconsistency and, if possible, provide a mechanistic explanation.

---

## [Referee Report · Reviewer #2 (Public review)]

Summary:

The authors exposed naïve male flies to different groups of females, either mated or virgin. Male flies can successfully copulate with virgin females; however, they are rejected by mated females. This rejection reduces sugar preference and sensitivity in males. Investigating the underlying neural circuits, the authors show that dopamine signaling onto GR5a sensory neurons is required for reduced sugar preference. GR5a sensory neurons respond less to sugar exposure when they lack dopamine receptors.

Strengths:

The findings add another strong phenotype to the existing dataset about brain-wide neuromodulatory effects of mating. The authors use several state-of-the-art methods, such as activity-dependent GRASP, to decipher the underlying neural circuitry. They further perform rigorous behavioral tests and provide convincing evidence for the local labellar circuit.

Weaknesses:

The authors focus on the circuit connection between dopamine and gustatory sensory neurons in the male SEZ. Therefore, it is still unknown how mating modulates dopamine signaling and what possible implications on other behaviors might result from a reduced sugar preference.

The authors updated missing literature in the manuscript and performed additional experiments regarding behavior, but also to further prove the functional connectivity between TH neurons and GR5a neurons.

I have no further recommendations.

---

## [Referee Report · Reviewer #3 (Public review)]

Summary

This study by Wang et al. explores a compelling link between two fundamental innate behaviors in *Drosophila melanogaster*, mating and feeding, demonstrating that repeated sexual failure in male flies leads to a transient yet reversible decrease in sweet taste perception. The authors show that this modulation is mediated by dopamine signaling from a specific subset of dopaminergic neurons in the subesophageal zone (SEZ) that directly influence Gr5a⁺ sweet-sensing neurons.

Aims of the Study

The authors aimed to understand whether unsuccessful mating attempts could affect sensory processing of sweet stimuli and thus feeding behavior in male fruit flies. They further sought to dissect the neural circuitry and molecular pathways underlying this behavioral plasticity, with a particular focus on dopaminergic modulation.

Major Strengths and Weaknesses

Strengths:

Novelty: The idea that reproductive experience modulates gustatory perception adds a new dimension to our understanding of cross-modal behavioral integration.Experimental approach: The study uses a broad array of genetic, pharmacological, imaging, and behavioral assays to demonstrate a causal relationship between sexual failure and reduced sweet perception, mediated by specific dopaminergic pathways.Methodological design: The authors link behavioral outcomes (reduced proboscis extension reflex) with neural activity (calcium imaging of Gr5a⁺ neurons) and molecular specificity (dopamine receptor subtype roles), providing a robust multi-level framework.

Weaknesses:

Ecological relevance: While the laboratory conditions are well controlled, the adaptive value or natural context of this taste modulation following mating failure remains speculative.

Achievement of Aims and Support for Conclusions

The authors have convincingly achieved their central aim. The results support the conclusion that sexual failure reduces sweet taste sensitivity through dopamine signaling. The reduced activity in Gr5a⁺ neuron after courtship rejection, its rescue by dopamine or successful copulation, and the requirement of specific dopamine receptors support the proposed model.

Impact and Utility

This work advances the field's understanding of how motivational states shaped by social experiences can directly influence sensory perception and behavior. It underscores the role of the dopaminergic system not only in reward but in integrating internal states across distinct behavioral responses. The experimental approach, including courtship conditioning paradigms and in vivo imaging methods, provides a valuable foundation for related studies in sensory modulation and behavioral plasticity.

Additional Context

This study supports a growing body of literature suggesting that insects possess emotion-like internal states that influence their behavior across contexts. The findings resonate with prior work on how stressors like social isolation or courtship failure lead to compensatory changes in other reward-seeking behaviors (e.g., ethanol consumption). Moreover, the concept that neural systems underlying basic drives like hunger and mating are dynamically interconnected may be conserved across phyla, suggesting broader relevance to understanding internal state-dependent modulation of behavior.

The authors addressed all the comments of previous reviews. The changes increased the clarity of the manuscript, the interpretation of the results and reinforce the conclusion.

---

## [Author Response]

The following is the authors’ response to the original reviews.

**Reviewer #1 (Public review):**
Wang et al. investigated how sexual failure influences sweet taste perception in male *Drosophila*. The study revealed that courtship failure leads to decreased sweet sensitivity and feeding behavior via dopaminergic signaling. Specifically, the authors identified a group of dopaminergic neurons projecting to the suboesophageal zone that interacts with sweet-sensing Gr5a+ neurons. These dopaminergic neurons positively regulate the sweet sensitivity of Gr5a+ neurons via DopR1 and Dop2R receptors. Sexual failure diminishes the activity of these dopaminergic neurons, leading to reduced sweet-taste sensitivity and sugar-feeding behavior in male flies. These findings highlight the role of dopaminergic neurons in integrating reproductive experiences to modulate appetitive sensory responses.Previous studies have explored the dopaminergic-to-Gr5a+ neuronal pathways in regulating sugar feeding under hunger conditions. Starvation has been shown to increase dopamine release from a subset of TH-GAL4 labeled neurons, known as TH-VUM, in the suboesophageal zone. This enhanced dopamine release activates dopamine receptors in Gr5a+ neurons, heightening their sensitivity to sugar and promoting sucrose acceptance in flies. Since the function of the dopaminergic-to-Gr5a+ circuit motif has been well established, the primary contribution of Wang et al. is to show that mating failure in male flies can also engage this circuit to modulate sugar-feeding behavior. This contribution is valuable because it highlights the role of dopaminergic neurons in integrating diverse internal state signals to inform behavioral decisions.An intriguing discrepancy between Wang et al. and earlier studies lies in the involvement of dopamine receptors in Gr5a+ neurons. Prior research has shown that Dop2R and DopEcR, but not DopR1, mediate starvation-induced enhancement of sugar sensitivity in Gr5a+ neurons. In contrast, Wang et al. found that DopR1 and Dop2R, but not DopEcR, are involved in the sexual failure-induced decrease in sugar sensitivity in these neurons. I wish the authors had further explored or discussed this discrepancy, as it is unclear how dopamine release selectively engages different receptors to modulate neuronal sensitivity in a context-dependent manner.

Our immunostaining experiments showed that three dopamine receptors, Dop1R1, Dop2R, and DopEcR were expressed in Gr5a^+^ neurons in the proboscis, which was consistent with previous findings by using RT-PCR (Inagaki et al 2012). As the reviewer pointed out, we found that Dop1R1 and Dop2R were required for courtship failure-induced suppression of sugar sensitivity, whereas Marella et al 2012 and Inagaki et al 2012 found that Dop2R and DopEcR were required for starvation-induced enhancement of sugar sensitivity. These results may suggest that different internal states (courtship failure vs. starvation) modulate the peripheral sensory system via different signaling pathways (e.g. different subsets of dopaminergic neurons; different dopamine release mechanisms; and different dopamine receptors). We have discussed these possibilities in the revised manuscript.

The data presented by Wang et al. are solid and effectively support their conclusions. However, certain aspects of their experimental design, data analysis, and interpretation warrant further review, as outlined below.(1) The authors did not explicitly indicate the feeding status of the flies, but it appears they were not starved. However, the naive and satisfied flies in this study displayed high feeding and PER baselines, similar to those observed in starved flies in other studies. This raises the concern that sexually failed flies may have consumed additional food during the 4.5-hour conditioning period, potentially lowering their baseline hunger levels and subsequently reducing PER responses. This alternative explanation is worth considering, as an earlier study demonstrated that sexually deprived males consumed more alcohol, and both alcohol and food are known rewards for flies. To address this concern, the authors could remove food during the conditioning phase to rule out its influence on the results.

This is an important consideration. To rule out potential confound from food intake during courtship conditioning, we have now also conducted courtship conditioning in vials absent of food. In the absence of any feeding opportunity over the 4.5-hour courtship conditioning period, sexually rejected males still exhibited a robust decrease in sweet taste sensitivity compared with Naïve and Satisfied controls (Figure 1-supplement 1C). These data confirm that the suppression of PER is driven by courtship failure per se, rather than by differences in feeding during the conditioning phase.

(2) Figure 1B reveals that approximately half of the males in the Failed group did not consume sucrose yet Figure 1-S1A suggests that the total volume consumed remained unchanged. Were the flies that did not consume sucrose omitted from the dataset presented in Figure 1-S1A? If so, does this imply that only half of the male flies experience sexual failure, or that sexual failure affects only half of males while the others remain unaffected? The authors should clarify this point.

Our initial description of the experimental setup might be a bit confusing. Here is a brief clarification of our experimental design and we have further clarified the details in the revised manuscript, which should resolve the reviewer’s concerns:

After the behavioral conditioning, male flies were divided for two assays. On the one hand, we quantified PER responses of individual flies. As shown in Figure 1C, Failed males exhibited decreased sweet sensitivity (as demonstrated by the right shift of the dose-response curve). On the other hand, we sought to quantify food consumption of individual flies by using the MAFE assay (Qi et al 2005).

In the initial submission, we used 400 mM sucrose for the MAFE assay. When presented with 400 mM sucrose, approximately 100% of the flies in the Naïve and Satisfied groups, and 50% of the flies in the Failed group, extended their proboscis and started feeding, as a natural consequence of decreased sugar sensitivity (Figure 1B). We were able to quantify the actual volume of food consumed of these flies showing PER responses towards 400 mM sucrose and observed no change (Figure 1-supplement 1A, left). To avoid potential confusion, we have now repeated the MAFE assay with 800 mM sucrose, which elicited feeding in ~100% of flies among all three groups, as shown in Figure 1C. Again, we observed no change in food intake (Figure 1-supplement 1A, right).

These experiments in combination suggest that sexual failure suppresses sweet sensitivity of the Failed males. Meanwhile, as long as they still responded to a certain food stimulus and initiated feeding, the volume of food consumption remained unchanged. These results led us to focus on the modulatory effect of sexual failure on the sensory system, the main topic of this present study.

(3) The evidence linking TH-GAL4 labeled dopaminergic neurons to reduced sugar sensitivity in Gr5a+ neurons in sexually failed males could be further strengthened. Ideally, the authors would have activated TH-GAL4 neurons and observed whether this restored GCaMP responses in Gr5a+ neurons in sexually failed males. Instead, the authors performed a less direct experiment, shown in Figures 3-S1C and D. The manuscript does not describe the condition of the flies used in this experiment, but it appears that they were not sexually conditioned. I have two concerns with this experiment. First, no statistical analysis was provided to support the enhancement of sucrose responses following activation of TH-GAL4 neurons. Second, without performing this experiment in sexually failed males, the authors lack direct evidence to confirm that the dampened response of Gr5a+ neurons to sucrose results from decreased activity in TH-GAL4 neurons.

We have now quantified the effect of TH^+^ neuron activation on Gr5a^+^ neuron calcium responses. in Naïve males, dTRPA1-mediated activation of TH^+^ cells significantly enhanced sucrose-induced calcium responses (Figure 3-supplement 1C); while in Failed males, the baseline activity of Gr5a^+^ neurons was lower (Figure 3C), the same activation also produced significant (even slightly larger) effect on the calcium responses of Gr5a^+^ neurons (Figure 3-supplement 1D).

Taken together, we would argue that these experiments using both Naïve and Failed males were adequate to show a functional link between TH^+^ neurons and Gr5a^+^ neurons. Combining with the results that these neurons form active synapses (Figure 3-supplement 1B) and that the activity of TH^+^ neurons was dampened in sexually failed males (Figure 3G-I), our data support the notion that sexual failure suppresses sweet sensitivity via TH-Gr5a circuitry.

(4) The statistical methods used in this study are poorly described, making it unclear which method was used for each experiment. I suggest that the authors include a clear description of the statistical methods used for each experiment in the figure legends. Furthermore, as I have pointed out, there is a lack of statistical comparisons in Figures 3-S1C and D, a similar problem exists for Figures 6E and F.

We have added detailed information of statistical analysis in each figure legend.

(5) The experiments in Figure 5 lack specificity. The target neurons in this study are Gr5a+ neurons, which are directly involved in sugar sensing. However, the authors used the less specific Dop1R1- and Dop2R-GAL4 lines for their manipulations. Using Gr5a-GAL4 to specifically target Gr5a+ neurons would provide greater precision and ensure that the observed effects are directly attributable to the modulation of Gr5a+ neurons, rather than being influenced by potential off-target effects from other neuronal populations expressing these dopamine receptors.

We agree with the reviewer that manipulating Dop1R1 and Dop2R genes (Figure 4) and the neurons expressing them (Figure 5) might have broader impacts. For specificity, we have also tested the role of Dop1R1 and Dop2R in Gr5a^+^ neurons by RNAi experiments (Figure 6). As shown by both behavioral and calcium imaging experiments, knocking down Dop1R1 and Dop2R in Gr5a^+^ neurons both eliminated the effect of sexual failure to dampen sweet sensitivity, further confirming the role of these two receptors in Gr5a^+^ neurons.

(6) I found the results presented in Fig. 6F puzzling. The knockdown of Dop2R in Gr5a+ neurons would be expected to decrease sucrose responses in naive and satisfied flies, given the role of Dop2R in enhancing sweet sensitivity. However, the figure shows an apparent increase in responses across all three groups, which contradicts this expectation. The authors may want to provide an explanation for this unexpected result.

We agree that there might be some potential discrepancies. We have now addressed the issues by re-conducting these calcium imaging experiments again with a head-to-head comparison with the controls (Gr5a-GCaMP, +/- Dop1R1 and Dop2R RNAi).

In these new experiments, Dop1R1 or Dop2R knockdown completely prevented the suppression of Gr5a^+^ neuron responsiveness by courtship failure (Figure 6E), whereas the activities of Gr5a^+^ neurons in Naïve/Satisfied groups were not altered. These results demonstrate that Dop1R1 and Dop2R are specifically required to mediate the decrease in sweet sensitivity following courtship failure.

(7) In several instances in the manuscript, the authors described the effects of silencing dopamine signaling pathways or knocking down dopamine receptors in Gr5a neurons with phrases such as 'no longer exhibited reduced sweet sensitivity' (e.g., L269 and L288), 'prevent the reduction of sweet sensitivity' (e.g., L292), or 'this suppression was reversed' (e.g. L299). I found these descriptions misleading, as they suggest that sweet sensitivity in naive and satisfied groups remains normal while the reduction in failed flies is specifically prevented or reversed. However, this is not the case. The data indicate that these manipulations result in an overall decrease in sweet sensitivity across all groups, such that a further reduction in failed flies is not observed. I recommend revising these descriptions to accurately reflect the observed phenotypes and avoid any confusion regarding the effects of these manipulations.

We have changed the wording in the revised manuscript. In brief, we think that these manipulations have two consequences: suppressing the overall sweet sensitivity, and eliminating the effect of sexual failure on sweet sensitivity.

**Reviewer #2 (Public review):**
Summary:The authors exposed naïve male flies to different groups of females, either mated or virgin. Male flies can successfully copulate with virgin females; however, they are rejected by mated females. This rejection reduces sugar preference and sensitivity in males. Investigating the underlying neural circuits, the authors show that dopamine signaling onto GR5a sensory neurons is required for reduced sugar preference. GR5a sensory neurons respond less to sugar exposure when they lack dopamine receptors.Strengths:The findings add another strong phenotype to the existing dataset about brain-wide neuromodulatory effects of mating. The authors use several state-of-the-art methods, such as activity-dependent GRASP to decipher the underlying neural circuitry. They further perform rigorous behavioral tests and provide convincing evidence for the local labellar circuit.Weaknesses:The authors focus on the circuit connection between dopamine and gustatory sensory neurons in the male SEZ. Therefore, it is still unknown how mating modulates dopamine signaling and what possible implications on other behaviors might result from a reduced sugar preference.

We agree with the reviewer that in the current study, we did not examine the exact mechanism of how mating experience suppressed the activity of dopaminergic neurons in the SEZ. The current study mainly focused on the behavioral characterization (sexual failure suppresses sweet sensitivity) and the downstream mechanism (TH-Gr5a pathway). We think that examining the upstream modulatory mechanism may be more suitable for a separate future study.

We believe that a sustained reduction in sweet sensitivity (not limited to sucrose but extend to other sweet compounds Figure 1-supplement 1D-E) upon courtship failure suggests a generalized and sustained consequence on reward-related behaviors. Sexual failure may thus resemble a state of “primitive emotion” in fruit flies. We have further discussed this possibility in the revised manuscript.

**Reviewer #3 (Public review):**
SummaryIn this work, the authors asked how mating experience impacts reward perception and processing. For this, they employ fruit flies as a model, with a combination of behavioral, immunostaining, and live calcium imaging approaches.Their study allowed them to demonstrate that courtship failure decreases the fraction of flies motivated to eat sweet compounds, revealing a link between reproductive stress and reward-related behaviors. This effect is mediated by a small group of dopaminergic neurons projecting to the SEZ. After courtship failure, these dopaminergic neurons exhibit reduced activity, leading to decreased Gr5a+ neuron activity via Dop1R1 and Dop2R signaling, and leading to reduced sweet sensitivity. The authors therefore showed how mating failure influences broader behavioral outputs through suppression of the dopamine-mediated reward system and underscores the interactions between reproductive and reward pathways.ConcernMy main concern regarding this study lies in the way the authors chose to present their results. If I understood correctly, they provided evidence that mating failure induces a decrease in the fraction of flies exhibiting PER. However, they also showed that food consumption was not affected (Fig. 1, supplement), suggesting that individuals who did eat consumed more. This raises questions about the analysis and interpretation of the results. Should we consider the group as a whole, with a reduced sensitivity to sweetness, or should we focus on individuals, with each one eating more? I am also concerned about how this could influence the results obtained using live imaging approaches, as the flies being imaged might or might not have been motivated to eat during the feeding assays. I would like the authors to clarify their choice of analysis and discuss this critical point, as the interpretation of the results could potentially be the opposite of what is presented in the manuscript.

Please refer to our responses to the Public Review (Reviewer 1, Point 2) for details.

**Recommendations for the authors:**

**Reviewer #1 (Recommendations for the authors):**
(1) The label for the y-axis in Figure 1B should be "fraction", not "percentage".

We have revised the figure as suggested.

(2) I suggest that the authors indicate the ROIs they used to quantify the signal intensity in Figure 3E and G.

We have revised the figures as suggested.

(3) There is a typo in Figure 4A: it should be "Wilde type", not "Wide type".

We have revised the figure as suggested.

(4) The elav-GAL4/+ data in Figure 4-S1B, C, and D appears to be reused across these panels. However, the number of asterisks indicating significance in the MAT plots differs between them (three in panels B and C, and four in panel D). Is this a typo?

It is indeed a typo, and we have revised the figure accordingly.

**Reviewer #2 (Recommendations for the authors):**
Additional comments:The authors should add this missing literature about dopamine and neuromodulation in courtship:Boehm et al., 2022 (eLife) - this study shows that mating affects olfactory behavior in females.Cazalé-Debat et al., 2024 (Nature) - Mating proximity blinds threat perception.Gautham et al., 2024 (Nature) - A dopamine-gated learning circuit underpins reproductive state-dependent odor preference in *Drosophila* females.

We have added these references in the introduction section.

Has the mating behavior been quantified? How often did males copulate with mated and virgin females?

We tried to examine the copulation behavior based on our video recordings. In the “Failed” group (males paired with mated females), we observed virtually no successful copulation events at all, confirming that nearly 100% of those males experienced sexual failure. In contrast, males in the “Satisfied” group (paired with virgin females) mated on average 2-3 times during the 4.5-hour conditioning period. We have added some explanations in the manuscript.

Do the rejected males live shorter? Is the effect also visible when they are fed with normal fly food, or is it only working with sugar?

We did not directly measure the lifespan of these males. But we conducted a relevant assay (starvation resistance), in which “Failed” males died significantly faster than both Naïve and Satisfied controls, indicating a clear reduction in their ability to endure food deprivation (Figure 1-supplement 1B). Since sweet taste is a primary cue for food detection in *Drosophila*, and sugar makes up a large portion of their standard diet, the drop in sugar sensitivity we observed in Failed males could likewise impair their perception and consumption of regular fly food, hence their resistance to starvation.

Also, the authors mention that the reward pathway is affected, this is probably the case as sugar sensation is impaired. One interesting experiment would be (and maybe has been done?) to test rejected males in normal odor-fructose conditioning. The data would suggest that they would do worse.

We have already measured how courtship failure affected fructose sensitivity (Figure 1 supplement 1D), and we found that the reduction in fructose perception was even more profound than for sucrose. We have not yet tested whether Failed males showed deficits in odor-fructose associative conditioning. That was indeed a very interesting direction to explore. But olfactory reward learning relies on molecular and circuit mechanisms distinct from those governing taste. We therefore argue such experiments would be more suitable in a separate, follow up study.

The authors could have added another group where males are exposed to other males. It would be interesting if this is also a "stressful" context and if it would also reduce sugar preference - probably beyond the scope of this paper.

In our experiments, all flies, including those in the Naïve, Failed, and Satisfied groups, were housed in groups of 25 males per vial before the conditioning period (and the Naïve group remained in the same group housing until PER testing). This means every cohort experienced the same level of “social stress” from male-male interactions. While it would indeed be interesting to compare that to solitary housing or other male-only exposures, isolation itself imposes a different kind of stress, and disentangling these effects on sugar preference would require a separate, dedicated study beyond the scope of the present work.

Would the behavior effect also show up with experienced males? Maybe this has been tested before. Does mating rejection in formerly successful males have the same impact?

As suggested by the reviewer, we performed an additional experiment in which males that had previously mated successfully were subsequently subjected to courtship rejection. As shown in Figure 1 supplement 1F, prior successful mating did not prevent the decline in sweet sensitivity induced by subsequent mating failure, indicating that even experienced males exhibit the reduction in sugar sensitivity after rejection.

Is the same circuit present and functioning in females? Does manipulating dopamine receptors in GR5a neurons in females lead to the same phenotype? This would suggest that different internal states in males and females could lead to the same phenotype and circuit modulations.

This is indeed a very interesting suggestion. In male flies, Gr5a-specific knockdown of dopamine receptors did not alter baseline sweet sensitivity, but it selectively prevented the reduction in sugar perception that followed mating failure (Figure 6C-D), indicating that this dopaminergic pathway is engaged only in the context of courtship rejection. By extension, knocking down the same receptors in female GR5a neurons would likewise be expected to leave their basal sugar sensitivity unchanged. Moreover, because there is currently no established paradigm for inducing mating failure in female flies, we cannot yet test whether sexual rejection similarly modulates sweet taste in females, or whether it operates via the same circuit.

**Reviewer #3 (Recommendations for the authors):**
Suggestions to the authors:Introduction, line 61. I suggest the authors add references in fruit flies concerning the rewarding nature of mating. For example, the paper from Zhang et al, 2016 "Dopaminergic Circuitry Underlying Mating Drive" demonstrates the role of the dopamine rewarding system in mating drive. There is a large body of literature showing the link between dopamine and mating.

We have added this literature in the introduction section.

Figure 1B and Figure Supplement 1: If I understood correctly, Figure Supplement 1A shows that the total food consumption across all tested flies remains unchanged. However, fewer flies that failed to mate consumed sucrose. I would be curious to see the results for sucrose consumption per individual fly that did eat. According to their results, individual flies that failed to mate should consume more sucrose. This would change the conclusion. The authors currently show that a group of flies that failed to mate consumed less sucrose overall, but since fewer males actually ate, those that failed to mate and did eat consumed more sucrose. The authors should distinguish between failed and satisfied flies in two groups: those that ate and those that did not.

Please see our responses to the Public Review for details (Reviewer 1, Point 2).

Figure 1C, right: For a better understanding of all the "MAT" figures, I suggest the authors start the Y axis with the unit 25 and increase it to 400. This would match better the text (line 114) saying that it was significantly elevated in the failed group. As it is, we have the impression of a decrease in the graph.

We have revised the figures accordingly.

Line 103: When suggesting a reduced likelihood of meal initiation of these males, do these males take longer to eat when they did it? In other words, is the latency to eat increased in failed males? That would be a good measure of motivational state.

We tried to analyze feeding latency in the MAFE assay by measuring the time from sucrose presentation to the first proboscis extension, but it was too short to be accurately accounted. Nevertheless, when conducting the experiments, we did not feel/observe any significant difference in the feeding latency between Failed males and Naïve or Satisfied controls.

Line 117. I don't understand which results the authors refer to when writing "an overall elevation in the threshold to initiate feeding upon appetitive cues". Please specify.

This phrase refers to the fact that for every sweet tastant we tested, including sucrose (Figure 1C), fructose and glucose (Figure 1 supplement 1D-E), the concentration-response curve in Failed males shifted to the right, and the Mean Acceptance Threshold (MAT) was significantly higher. In other words, for these different appetitive cues, mating failure raised the concentration of sugar required to trigger a proboscis extension, indicating a general elevation in the threshold to initiate feeding upon an appetitive cue.

Figure 1D. Please specify the time for the satisfied group.

For clarity, the Naïve and Satisfied groups in Figure 1D each represent pooled data from 0 to 72 hours post-treatment, as their sweet sensitivity remained stable throughout this period. Only the Failed group was shown with time-resolved data, since it was the only group exhibiting a dynamic change in sugar sensitivity over time. We have now specified this in the figure legend.

Figure 1F. The phenotype was not totally reversed in failed-re-copulated males. Could it be due to the timing between failure and re-copulation? I suggest the authors mention in the figure or in the text, the time interval between failure and re-copulation.

We’d like to clarify that the interval between the initial treatment (“Failed”) and the opportunity for re copulation was within 30 minutes. The incomplete reversal in the Failed-re-copulated group indeed raised interesting questions. One possible explanation is that mating failure reduces synaptic transmissions between the SEZ dopaminergic neurons and Gr5a^+^ sweet sensory neurons (Figure 3), and the regeneration of these transmissions takes a longer time. We have added this information to the figure legend and the Method section.

Line 227-228 and Figure 3E. The authors showed that the synaptic connections between dopaminergic neurons and Gr5a+ GRNs were significantly weakened. I am wondering about the delay between mating failure and the GFP observation. It would be informative to know this timing to interpret this decrease in synaptic connections. If the timing is relatively long, it is possible that we can observe a neuronal plasticity. However, if this timing is very short, I would not expect such synaptic plasticity.

The interval between the behavioral treatment and the GRASP-GFP experiment was approximately 20 hours. We chose this time window because it was sufficient for both GFP expression and accumulation. Therefore, the observed reduction in synaptic connections between dopaminergic neurons and Gr5a^+^ GRNs likely reflects a genuine, experience-induced structural and functional change rather than an immediate, transient effect. We have added this information to the revised manuscript for clarity in the Method section.

Line 240-243: The authors demonstrated that there is a reduction of CaLexA-mediated GFP signals in dopaminergic neurons in the SEZ after mating failure, but not a reduction in Gr5a+ GRNs. I suggest replacing "indicate" with "suggest' in line 240.

We have made the change accordingly. Meanwhile, we would like to clarify that while we observed a reduction of NFAT signal in SEZ dopaminergic neurons (Figure 3G), we did not directly test NFAT signal in Gr5a^+^ neurons. Notably, the results that the synaptic transmissions from SEZ dopaminergic neurons to Gr5a^+^ neurons were weakened (Figure 3E-F), and the reduction of NFAT signal in SEZ dopaminergic neurons (Figure 3G-I), were in line with a reduction in sweet sensitivity of Gr5a^+^ neurons upon courtship failure (Figure 3B-D).

Line 243: replace "consecutive" with "constitutive".

We have revised it accordingly.

Figure 5: I have trouble understanding the results obtained in Figure 5. Both constitutive activation and inhibition of Dop1R1 and Dop2R neurons lead to the same results, knowing that males who failed mating no longer exhibit decreased sweet sensitivity. I would have expected contrary results for both experimental conditions. I suggest the author to discuss their results.

Both activation and inhibition of Dop1R1 and Dop2R neurons eliminated the effect of courtship failure on sweet sensitivity (Figure 5). These results are in line with our hypothesis that courtship failure leads to changes in dopamine signaling and hence sweet sensitivity. If dopamine signaling via Dop1R1 and Dop2R was locked, either to a silenced or a constitutively activated state, the effect of courtship failure on sweet sensitivity was eliminated.

Nevertheless, as the reviewer pointed out, constitutive activation/inhibition should in principle lead to the opposite effect on Naïve flies. In fact, when Dop1R1^+^/Dop2R^+^ neurons were silenced in Naïve flies, PER to sucrose was significantly reduced (Figure 5C-D), confirming that these neurons normally facilitate sweet sensation. Meanwhile, while neuronal activation by NaChBac did show a trend towards enhanced PER compared to the GAL4/+ controls, it did not exhibit a difference compared to +>UAS-NaChBac controls that showed a high PER level, likely due to a potential ceiling effect. We have added the discussions to the manuscript.

Figure 7: I suggest the authors modify their figure a bit. It is not clear why in failed mating, the red arrow in "behavioral modulation" goes to the fly. The authors should find another way to show that mating failure decreased the percentage of flies that are motivated to eat sugar.

We have modified the figure as suggested.

Overall, I would suggest the authors be precautious with their conclusion. For example, line 337 = "sexual failure suppressed feeding behavior". This is not what is shown by this study. Here, the study shows that mating failure decreases the fraction of flies to eat sucrose. Unless the authors demonstrate that this decrease is generalizable to other metabolites, I suggest the authors modify their conclusion.

While we primarily used sucrose as the stimulant in our experiments, we also tested responses to two other sugars: fructose and glucose (Figure 1 supplement 1D-E). In all three cases, mating failure led to a significant reduction in sweet perception, suggesting that the effect of courtship failure is not limited to a single metabolite but rather reflects a general decrease in sweet sensitivity. Meanwhile, reduced sweet sensitivity indeed led to a reduction of feeding initiation (Figure 1).